
# Quantum robustness and phase transitions
# of the 3D toric code in a field

**David A. Reiss** [1,2⋆] **and Kai Phillip Schmidt** [2†]

**1** Dahlem Center for Complex Quantum Systems and Physics Department,
Freie Universität Berlin, Arnimallee 14, 14195 Berlin, Germany
**2** Chair for Theoretical Physics 1, Universität Erlangen-Nürnberg,
Staudtstraße 7, 91058 Erlangen, Germany

⋆ david.reiss@fu-berlin.de, † kai.phillip.schmidt@fau.de

## Abstract

We study the robustness of 3D intrinsic topological order under external perturbations by investigating the paradigmatic microscopic model, the 3D toric code in an external magnetic field. Exact dualities as well as variational calculations reveal a ground-state phase diagram with first and second-order quantum phase transitions. The variational approach can be applied without further approximations only for certain field directions. In the general field case, an approximative scheme based on an expansion of the variational energy in orders of the variational parameters is developed. For the breakdown of the 3D intrinsic topological order, it is found that the (im-)mobility of the quasiparticle excitations is crucial in contrast to their fractional statistics.


# 1   Introduction

The search for undiscovered quantum facets of nature is one of the most active and fascinating lines of research in modern physics, both from the perspective of fundamental research as well as technology. This is evident in strongly correlated quantum matter displaying intrinsic topological order [1–3]. Such quantum phases display intriguing quantum phenomena like long-range entanglement and degeneracy of the ground state, depending on the genus of the bulk topology. Additionally in two dimensions, they feature exotic point-like quasiparticles, so-called anyons [4, 5], having fractional particle statistics different from fermions or bosons. Therefore, the concept of intrinsic topological order carries on our understanding of nature's secrets beyond the theories of Landau and Goldstone, which are based on spontaneous symmetry breaking and dominated condensed matter physics for several decades. Furthermore, these physical properties are exploited in proposals to employ such phases as topological quantum memories or topological quantum computers [6, 7].

Experimentally accessible, intrinsic topological order causes the two-dimensional fractional quantum Hall effect at certain filling fractions [8, 9] with strongly correlated electronic degrees of freedom. Intrinsic topological order is also expected for quantum magnets in so-called quantum spin liquid phases [10, 11], which might be of importance for high-temperature superconductivity [12, 13]. In this context, Mott insulators with strong spin-orbit interaction like the layered iridates [14, 15] as well as $\alpha$-RuCl$_3$ [16–18] have been investigated intensely in recent years. These quantum materials are potentially approximate instances of the two-dimensional Kitaev's honeycomb model [19], which is known to possess topologically ordered ground states. Kitaev's honeycomb model features three different kinds of Ising interactions. When one kind of interaction is much stronger than the other two, the 2D toric code [6] arises perturbatively as an effective low-energy model in fourth-order of the two weak interaction strengths [19–22] while higher orders induce attractive interactions between its quasiparticle excitations, but do not alter the topological ordering [20–22].

The 2D toric code was proposed by Kitaev in 2003 as a topologically protected, self-correcting quantum memory. It represents an exactly solvable paradigmatic microscopic model for intrinsic topological order featuring all its relevant aspects like anyonic quasiparticle excitations. As a consequence, there have been many studies using the 2D toric code as starting point for investigating the physical properties of intrinsic topological order, e.g., its robustness

under external perturbations [23–34], the (in-)stability under thermal fluctuations [35–37], the properties of entanglement measures [38, 39], the calculation of dynamical correlation functions [40] as well as non-equilibrium properties [41, 42]. A first step towards experimentally implementing the 2D toric code and its exotic excitations was taken by the realization of the highly-entangled ground state of the 2D toric code in quantum simulators. These experiments were proposed for quantum simulators utilizing trapped ions, photons and NMR in 2007 [43]. 2D photonic experiments were conducted successfully in 2009 [44, 45], as well as 2D NMR experiments in 2007 [46] and 2012 [47]. Furthermore, it was proposed how to realize the toric code Hamiltonian in systems of ultracold atoms [48], polar molecules in optical lattices [49], and with lattices of superconducting circuits [50]. Further realizations of the 2D toric code Hamiltonian exist in NMR systems [51] as well as with laser-excited Rydberg atoms [52].

Much less is known for systems with intrinsic topological order in three dimensions. One major difference to 2D is the nature of the elementary excitations, since point-like excitations with exotic particle statistics are absent according to the spin-statistics theorem. However, apart from point-like bosonic or fermionic degrees of freedom, typically there exist extended excitations on loops or membranes with anyonic statistics. The 3D version of the toric code [53, 54] is a microscopic model which hosts such extended excitations but also point-like excitations having non-trivial mutual statistics. The toric code can in principle be realized in 3D quantum simulator setups, like particles in 3D optical lattices, 3D magnetic trap arrays or laser-excited Rydberg atoms [52]. Beside that there exist 3D versions of Mott insulators with strong spin-orbit interactions [55] for which the 3D toric code potentially emerges as an effective low-energy model from 3D Kitaev models analogous to the 2D case. Other but similar approaches to realize the 3D toric code have been suggested recently [56, 57]. Another different class of 3D topological order are so-called fracton phases [58–63], which have come into focus recently. One advantage of fracton phases are their potential stability against thermal fluctuations, in contrast to the instability of the toric code in lower than 4 dimensions [35, 36] which therefore cannot be used as a topologically-protected quantum memory in practice.

In this work we present a theoretical investigation of the robustness and quantum phase transitions of 3D intrinsic topological order under external perturbations, by taking the example of the 3D toric code. The main motivation is that its quasiparticles feature exotic mutual statistics of point-like and spatially extended loop-like quasiparticles [64] in contrast to their 2D point-like counterparts. More specifically, we explore the effect of the statistics on the robustness and quantum phase transitions of the 3D toric code, when the quasiparticles become dynamical due to an external homogeneous magnetic field. Overall, we find that the (im-)mobility of the quasiparticle excitations in contrast to their fractional statistics is crucial for the breakdown of the 3D intrinsic topological order. Furthermore, there are several other reasons which motivate the investigation of the perturbed 3D toric code from a theoretical perspective: i) the perturbing magnetic field induces generic quantum fluctuations, which are ubiquitous due to Heisenberg's uncertainty principle. ii) The unperturbed toric code is exactly soluble, can be classified in terms of tensor categories and can be explained physically by the mechanisms of string-net condensation [65]. iii) Its low-energy physics can be described by topological field theories [66], which are well understood in 2D, but far less in 3D. iv) The 2D as well as the 3D toric code can be described by different powerful mathematical theories [11, 66–69]. v) Like the 2D version, the 3D toric code represents the paradigmatic model for investigating physical properties of 3D intrinsic topological order.

The paper is organized as follows. In Sect. 2 we comprehensively review the properties of the unperturbed 3D toric code including the ground states and the nature of the elementary excitations. Furthermore, we discuss the leading effects of a homogeneous magnetic field on these excitations. Next the exact duality transformations for single-field cases are explained in

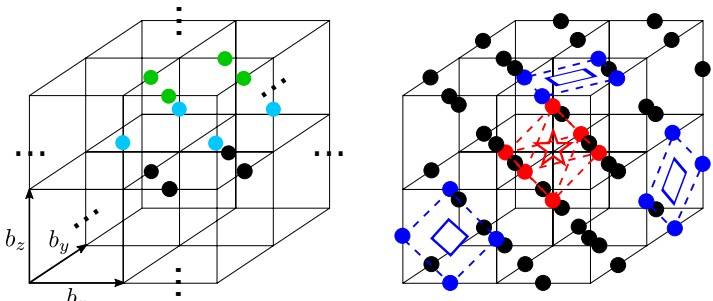

Figure 1: Lattice, degrees of freedom and interactions of the 3D toric code considered in this post. Left: the lattice is defined by the basis vectors $b_x, b_y$ and $b_z$. The spin $-1/2$ degrees of freedom are depicted as spheres located on the links of the lattice; for better visibility, only the spins of one elementary cube are shown. The color coding indicates the different x-y-planes and the dots the translationally invariant thermodynamic limit. Right: three kinds of plaquette operators depicted in blue and a star operator depicted in red, as described in the main text. For clarity only one star and one of each kind of plaquette operators are shown. The illustration is adapted from [71].

Sect. 3. In Sect. 4, all the technical aspects of the variational approaches to approximate the ground states are presented. The overall results for the quantum phase diagram are contained in Sect. 5. We conclude the work and give a short outlook in Sect. 6.

## 2   3D toric code and magnetic fields

In the following we first describe the properties of the unperturbed 3D toric code, which is not as known as its 2D counterpart [11, 70]. Afterwards we consider the 3D toric code in a uniform magnetic field and discuss leading effects of the field on the topological properties.

### 2.1   Unperturbed 3D toric code

The toric code can be defined for spins (qubits) located on the links of different lattices, e.g., the square lattice, the honeycomb lattice, as well as other trivalent lattices in 2D and 3D like those considered in [55]. All methods of this post can be applied to the toric code on different lattices. Here we consider the cubic lattice, see Fig. 1 left. The 3D toric code is defined by the Hamiltonian

$$H := -\frac{1}{2}\sum_{\substack{\text{stars}\\s}}A_s - \frac{1}{2}\sum_{\substack{\text{plaquettes}\\p}}B_p\,; \qquad A_s := \left(\prod_{j\in s}\sigma_j^x\right), \qquad B_p := \left(\prod_{j\in p}\sigma_j^z\right), \qquad (1)$$

where $\sigma_j^x, \sigma_j^y$ and $\sigma_j^z$ are the usual Pauli matrices acting on the spin $j$ of the system. The star operators $A_s$ act on the spins in a "star" $s$ around a vertex and plaquette operators $B_p$ on the spins in a plaquette $p$ of the lattice as shown in Fig. 1. The star and plaquette operators commute, i.e.,

$$[A_s, A_{s'}] = [B_p, B_{p'}] = [A_s, B_p] = 0 \quad \forall s, s', p, p', \qquad (2)$$

as the star and plaquette operators either act on none or two common spins. Consequently,

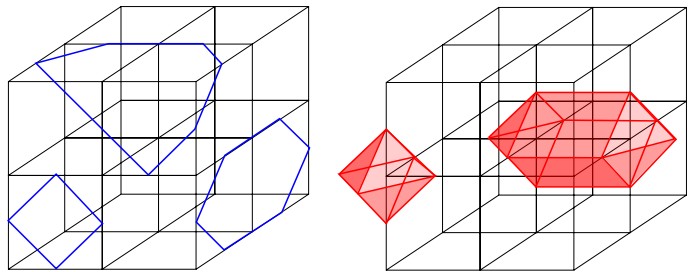

Figure 2: Loop soup and membrane soup picture of the ground state of the 3D toric code. Left: the spins located at crossings of the blue lines and the cubic lattice links are flipped with respect to the $\sigma^x$-basis. Analogously right, the spins located at the crossings of the red lines (corners of polyhedrons) and the cubic lattice links are flipped with respect to the $\sigma^z$-basis.

the eigenvalues of star and plaquette operators are conserved and equal to

$$a_s = \pm 1 \,, \ b_p = \pm 1 \quad \forall s, p \,, \tag{3}$$

due to the star and plaquette operators squaring to the identity. Thus the model is exactly soluble and the ground state is constrained by the condition $a_s = +1$, $b_p = +1 \ \forall s, p$. In the $\sigma^x$-basis ($\sigma^x \left| \rightarrow \right\rangle = \left| \rightarrow \right\rangle$, $\sigma^x \left| \leftarrow \right\rangle = -\left| \leftarrow \right\rangle$), this can be ensured by the construction

$$\prod_s \frac{\mathbb{1} + A_s}{2} \prod_p \frac{\mathbb{1} + B_p}{2} \left| \rightarrow \rightarrow \cdots \rightarrow \right\rangle = \prod_p \frac{\mathbb{1} + B_p}{2} \left| \rightarrow \rightarrow \cdots \rightarrow \right\rangle = $$
$$= \frac{1}{2^{3N}} \Big( \mathbb{1} + \sum_p B_p + \sum_{p, p' \neq p} B_p B_{p'} + \dots \Big) \left| \rightarrow \rightarrow \cdots \rightarrow \right\rangle \,, \tag{4}$$

where $N$ is the number of unit cells. This amounts to an equal-weight superposition of all states with loops of flipped spins and is a 3D generalization of the loop soup ground state of the 2D toric code. One state in this superposition is pictorially represented in the left part of Fig. 2. In contrast, in the picture of the $\sigma^z$-basis ($\sigma^z \left| \uparrow \right\rangle = \left| \uparrow \right\rangle$, $\sigma^z \left| \downarrow \right\rangle = -\left| \downarrow \right\rangle$), the same projection of the state $\left| \uparrow \uparrow \dots \uparrow \right\rangle$ results in a "membrane soup", as star operators flip spins on closed membranes, illustrated in Fig. 2 right. Equivalently, one could start with any other product state $\left| \vec{h} \vec{h} \dots \vec{h} \right\rangle$ where all spins point in the direction of a magnetic field $\vec{h}$.

**Ground-state entanglement and degeneracy** – Due to this structure of the ground state which contains arbitrarily large loops and membranes, it is long-range entangled and satisfies an area law for the entanglement entropy [72], modified by a universal non-trivial topological entanglement entropy of $\gamma = 2 \ln(2)$ [71], equal to that of the 2D toric code.[1] In contrast to $\gamma_{2D} = 0$ for the 2D case at $T > 0$, one has

$$\gamma = \begin{cases} 2\ln(2) & \text{for} \quad T = 0 \,, \\ \ln(2) & \text{for} \quad 0 < T \leq T_c = 1.313346(3)J \qquad (\text{here } J = \tfrac{1}{2}) \,, \\ 0 & \text{for} \quad T > T_c \,. \end{cases} \tag{5}$$

---

[1] The definitions of the topological entanglement entropy by [73] and [74] differ by a factor of 2; here the latter definition was chosen.

Table 1: Constraints of the 3D toric code as described in the main text; plane $\alpha \in \{xy, xz, yz\}$.

| name | # independent | form | reformulation |
|---|---|---|---|
| "volume" | 1 | $\prod\limits_{s} A_s = \mathbb{1}$ | $\Rightarrow \quad A_s = \prod\limits_{s', s' \neq s} A_{s'}$ |
| "cube" | $N-1$ | $\prod\limits_{p \in \text{cube } c} B_p = \mathbb{1}$ | $\Rightarrow \quad B_p = \prod\limits_{p' \in c, p' \neq p} B_{p'}$ |
| | (not $N$, as) | | $\prod\limits_{p \in c} B_p = \prod\limits_{c', c' \neq c} \prod\limits_{p \in c'} B_p$ |
| "plane" | 3 | $\prod\limits_{p \in \text{plane } \alpha} B_p = \mathbb{1}$ | $\Rightarrow \quad B_p = \prod\limits_{p' \in \alpha, p' \neq p} B_{p'}$ |
| | (not more, as) | | $\prod\limits_{p \in \alpha} B_p = \prod\limits_{p' \in c} B_{p'} \prod\limits_{p \in \alpha} B_p$, etc. |

This coincides with non-analyticities of the canonical partition function $Z$, which indicates *finite*-temperature phase transitions. Still, the 3D toric code with periodic boundary conditions (PBC) at finite temperature is not a model for a thermally-stable fault-tolerant quantum memory, because it can only store a probabilistic bit [37, 71].

At zero temperature, the 3D toric code features non-local, topologically-protected logical qubits, which will be shown in the following. The 3D toric code on the cubic lattice with $N$ cubes and PBC possesses $3N$ spins, $N$ stars, $3N$ plaquettes and the constraints listed in Tab. 1.[2] These constraints are illustrated in Fig. 3. The products of plaquette operators forming closed membranes equal the identity, too. But they are not independent of the $N-1$ "cube constraints", as they can be constructed from products of the latter. Similarly, only 3 of the "plane constraints" are independent, because the planes can be deformed by multiplication with cube constraints. Consequently, the ground-state degeneracy is

$$\frac{2^{N_{\text{spins}}}}{\dim(E_H)} = \frac{2^{N_{\text{spins}}}}{2^{N_{\text{stars}}+N_{\text{plaquettes}}-N_{\text{constraints}}}} = \frac{2^{3N}}{2^{N+3N-1-(N-1)-3}} = 2^3, \tag{6}$$

where $E_H$ denotes the eigenspace of the Hamiltonian. The different ground-state sectors can be discriminated by the conserved eigenvalues of three topologically different non-local, non-contractible, commuting closed membrane operators

$$W_\alpha^m := \prod_{j \in \mathcal{P}_\alpha^m} \sigma_j^x, \qquad [W_\alpha^m, A_s] = [W_\alpha^m, B_p] = 0 \quad \forall s, p, \tag{7}$$

where $\alpha \in \{xy, xz, yz\}$. The planes $\mathcal{P}_\alpha^m$ are defined as

$$\mathcal{P}_{xy}^m := \{j_{p,q} = \frac{1}{2}b_x + \frac{1}{2}p(b_x + b_y) + \frac{1}{2}q(b_x - b_y) + \left(n_z + \frac{1}{2}\right)b_z \,|\, p, q \in \mathbb{Z}\}, \tag{8}$$

with some arbitrary fixed $n_z \in \mathbb{Z}$ and the basis vectors $b_\beta$ in $\beta$-direction of Fig. 1, $\beta \in \{x, y, z\}$:

$$b_x = (1, 0, 0), \quad b_y = (0, 1, 0), \quad b_z = (0, 0, 1). \tag{9}$$

The planes $\mathcal{P}_{xz/yz}^m$ are constructed analogously. One such membrane operator is illustrated in the left part of Fig. 4. These operators measure the parity of the number of loops whose spins

---

[2]This depends on the considered lattice, but the counting approach can also be applied to other lattices. In the case of open boundary conditions, relevant for small-scale experimental implementations of the 3D toric code, the ratios of the number of spins, stars and plaquettes depend on the systems' sizes and kinds of boundaries, analogously to 2D called "smooth" and "rough" boundaries [75,76]. Only the $N-1$ closed-cell constraints survive. Like in 2D, it is possible to construct systems with a non-trivial ground-state degeneracy.

point left, winding in the direction perpendicular to the membrane around the 3-torus. In order to create such loops, one can employ a set of non-local, non-contractible loop operators

$$W_\beta^e := \prod_{j \in \mathcal{L}_\beta^e} \sigma_j^z; \qquad [W_\beta^e, A_s] = [W_\beta^e, B_p] = 0 \quad \forall s, p, \tag{10}$$

where $\beta \in \{x, y, z\}$ and the loops $\mathcal{L}_\beta^e$ are defined as

$$\mathcal{L}_x^e := \{j_n = \left(n + \frac{1}{2}\right) b_x + n_y b_y + n_z b_z \,|\, n \in \mathbb{Z}\}, \tag{11}$$

with some fixed $n_x, n_y \in \mathbb{Z}$, $\mathcal{L}_{y/z}^e$ analogously. The operators $W_\beta^e$ toggle between the different ground-state sectors, as

$$W_{x/y/z}^e W_{yz/xz/xy}^m = -W_{yz/xz/xy}^m W_{x/y/z}^e. \tag{12}$$

These three different kinds of loops and loop operators are illustrated in the right part of Fig. 4. In the membrane-soup picture of the $\sigma^z$-basis, the sets of operators $W_\alpha^m$ and $W_\beta^e$ change roles of discriminant and toggle between the different ground-state sectors. One can show in general that the ground-state degeneracy is $2^{b_1}$ on a manifold with first Betti number $b_1$, which is the number of topologically different non-contractible loops or membranes to each of which we can associate a loop/membrane operator as above. For example the 3D toric code on the *solid 2-torus* with a "smooth" boundary [75, 76] has one topologically different non-contractible loop and thus a two-fold degenerate ground state.

**Excitations** – If some eigenvalues of $A_s$ or $B_p$ equal $-1$, the 3D toric code is in an excited state.[3] The excitations, called $e$ for $a_s = -1$ and $m$ for $b_p = -1$, can be created by

$$L_{s,s'}^e := \prod_{j \in \mathcal{L}_{s,s'}^e} \sigma_j^z, \qquad M_{\partial \mathcal{P}}^m := \prod_{j \in \mathcal{P}^m} \sigma_j^x, \tag{13}$$

where the open string $\mathcal{L}_{s,s'}^e$ is defined as $\mathcal{L}_x^e$ by a composition of links, with end vertices $s$ and $s'$. The open membrane $\mathcal{P}^m$ is constructed out of faces whose midpoints are the spin locations and whose normal vectors are parallel to the respective link, as illustrated in Fig. 5. Like $e$-quasiparticles (QP) of the 2D toric code, the $e$-excitations, which are their own antiparticles, are created, moved and annihilated at the endpoints of the open string. They are hardcore-bosonic point particles. The smallest possible membranes, centered at only one spin $j$, define the operator

$$M_{\partial \mathcal{P}_j}^m := \sigma_j^x, \tag{14}$$

which, applied to the ground state, creates four $m$-excitations, as shown in the left part of Fig. 5. This configuration of excitations along the closed loop $\partial \mathcal{P}_j$ will be called 4-$m$-loop. In general, $m$-excitations can only be created, annihilated and moved in loops of $4, 6, 8$, and higher even numbers of $m$-excitations; thus it is appropriate to interpret $m$-excitations rather as spatially extended excitations than as point particles. For convenience, a single $m$-excitation will be called an $m$-quasiparticle, too. The creation of single $m$-excitations is impossible, but the physical wave function of a 1-$m$-state can be written down as

$$|m_p\rangle := \frac{\mathbb{1} - B_p}{2} \prod_{p', p' \neq p} \frac{\mathbb{1} + B_{p'}}{2} |\rightarrow \rightarrow \cdots \rightarrow\rangle. \tag{15}$$

---

[3]The following paragraph can also be viewed, in the light of quantum codes, as the dynamics of uncorrected errors.

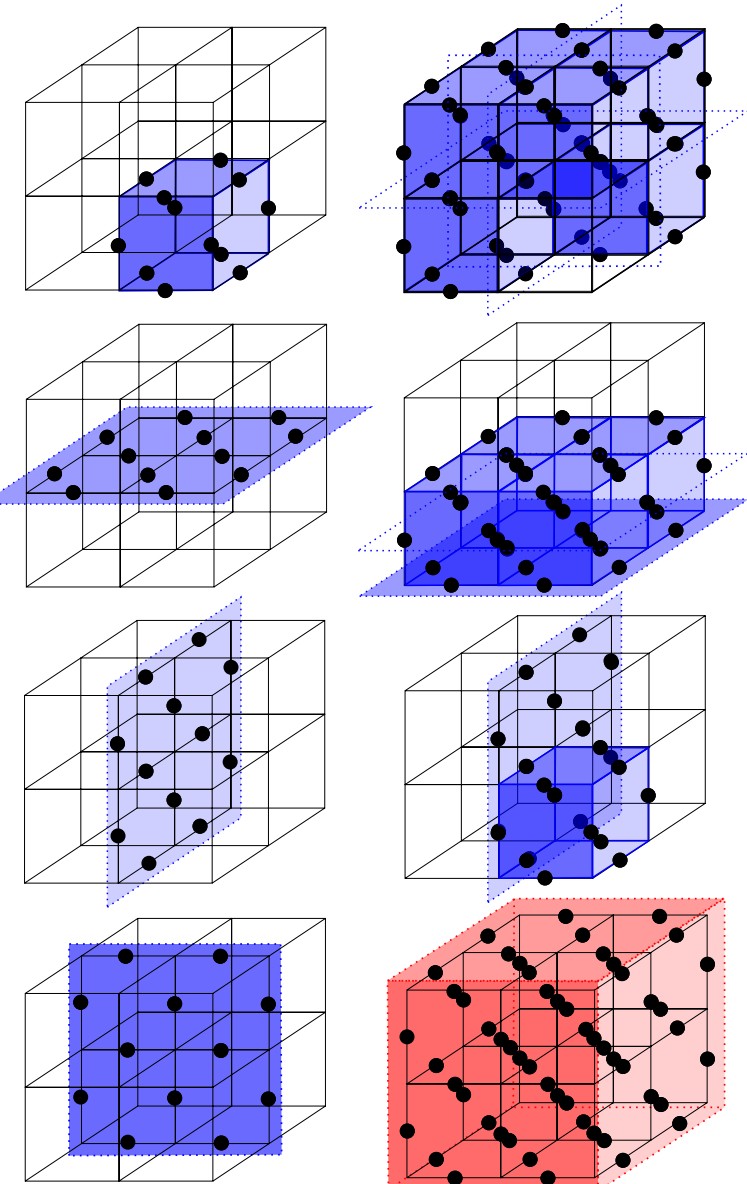

Figure 3: First line left: a cube constraint of the 3D toric code; right: equivalent constraint resulting from the product of all but one cube constraints. Second line left: x-y-plane constraint; right: equivalent constraint resulting from the product of another x-y-plane constraint and all adjacent cube constraints. Third line left: y-z-plane constraint; right: plane constraint deformed by multiplication with a cube constraint. Fourth line left: x-z-plane constraint; right: star constraint resulting from the product of all star operators. Dots indicate the translationally invariant continuation of the configuration.

This state belongs to a different superselection sector of the Hilbert space with respect to all local and non-local observables. There does not exist an analogon of the non-local operator $L^e_{s,\infty}$ (13) of an open string going to infinity, creating or annihilating a single $e$.

The precise form of the strings of $L^e_{s,s'}$ (membranes of $M^m_{\partial\mathcal{P}}$) above are irrelevant as (long as) we can deform them by application of $B_p$ with $b_p = +1$ ($A_s$ with $a_s = +1$). Moving an $e$-QP at $s$ via a suitable loop operator $L^e_{s,s}$ in a closed path through a loop of $m$-excitations or

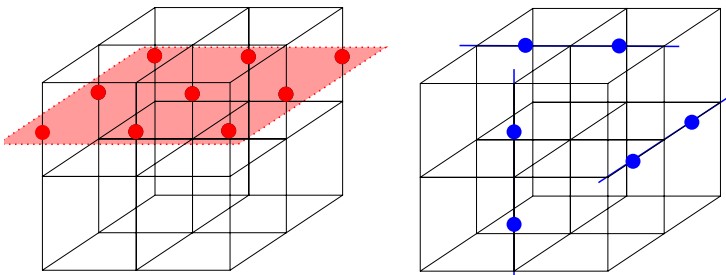

Figure 4: Non-contractible closed membrane operators $W_{xy}^m$ (left) and loop operators $W_x^e$, $W_y^e$, $W_z^e$ (right). For clarity only the involved spins of the red colored plane $\mathcal{P}_{xy}^m$ (left) and blue colored loops $\mathcal{L}_x^e$, $\mathcal{L}_y^e$, $\mathcal{L}_z^e$ (right) are shown, as well as only one kind of membrane operator. Red (blue) coloring of spheres means that Pauli matrices $\sigma_x$ ($\sigma_z$) are applied to the spins. Dots indicate that the membrane and loops span the whole system.

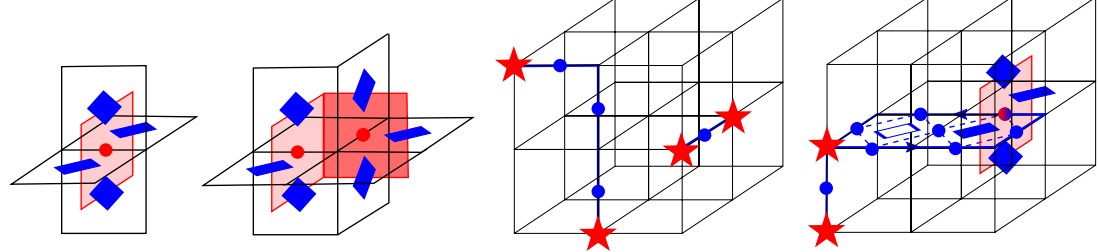

Figure 5: Left: open membrane operators $M_{\partial\mathcal{P}}^m$ (first and second left) and loop operators $L_{s,s'}^e$ (third left). For clarity only the involved spins and some parts of the cubic lattice are shown. Red (blue) coloring of spheres means that Pauli matrices $\sigma_x$ ($\sigma_z$) are applied to the spins. A filled parallelogram (star) indicates that an $m$-excitation ($e$-excitation) is present at the corresponding location. Right: exotic mutual statistics of a 4-$m$-loop (four filled blue plaquettes) and an $e$-QP (red star). The operator moving the $e$ along the closed blue loop equals the product of the two indicated plaquette operators, which yield a phase factor of $-1$, see Eq. (16).

moving a loop of $m$-excitations at $\partial\mathcal{P}$ via a suitable closed-membrane operator $M_{\partial\mathcal{P}'=\emptyset}^m$ in a closed path over an $e$-QP results in a phase factor of $-1$: closed-loop operators $L_{s,s}^e$ detect the presence of $m$-QP inside the loop and closed-membrane operators $M_{\partial\mathcal{P}'=\emptyset}^m$ detect the presence of $e$-QP inside the membrane, as

$$L_{s,s}^e = \prod_{\substack{p\in\mathcal{P}^e: \\ \partial\mathcal{P}^e=\mathcal{L}_{s,s}^e}} B_p\,, \qquad M_{\partial\mathcal{P}'=\emptyset}^m = \prod_{\substack{s\in\mathcal{V}^m: \\ \partial\mathcal{V}^m=\mathcal{P}'^m}} A_s\,, \tag{16}$$

where the membrane $\mathcal{P}^e$ consists of faces of the original cubic lattice and the volume $\mathcal{V}^m$ is constructed from elementary cubes centered around the stars (vertices) of the original cubic lattice. This is illustrated in the right part of Fig. 5. Consequently, if the wordline of $e$-QP and $m$-loop form a linked knot, a phase factor of $-1$ occurs; if they are unlinked, the phase factor is trivially $+1$. Alternatively, one can show this by the anticommutation of the Pauli matrices acting on the spin colored both red and blue in the right part of Fig. 5. This exotic mutual statistics of point and spatially extended particles emerging in a system of spins (hardcore bosons) is beyond bosonic or fermionic statistics in 3D. It is a macroscopic non-local quantum

effect: it even occurs when $e$-QP and $m$-loop are braided in a linked knot of macroscopic size.

In the quasiparticle picture introduced above, the non-contractible loop operators $W_\beta^e$ (10) discriminating (toggling) between the different ground states can be interpreted as creating a pair of $e$-quasiparticles, moving them in a non-contractible loop in $b_\beta$-direction around the torus and annihilating them again. This shows that ground-state degeneracy and deconfined anyonic excitations of phases with topological quantum order are interlinked.

Altogether, the 3D toric code shows all the signature properties of topological quantum order: topological ground-state degeneracy, absence of local order parameters, long-range, area-law entanglement entropy modified by a non-zero universal topological entanglement entropy and deconfined fractional excitations.

## 2.2 3D toric code in a uniform magnetic field

As the last subsection showed, the QP of the unperturbed toric code are static and non-interacting. However, due to perturbations and quantum fluctuations, the QP gain dynamics, become dressed and start to interact, which finally leads to the breakdown of the topological order for finite values of the perturbation. Here we consider the simplest possible perturbation of the 3D toric code in the form of a uniform magnetic field:

$$H_{\text{TCF}} = -\frac{1}{2}\sum_s A_s - \frac{1}{2}\sum_p B_p - \vec{h} \cdot \sum_{\text{spins } j} \vec{\sigma}_j, \tag{17}$$

where $\vec{\sigma}_j$ denotes the vector of Pauli matrices $\vec{\sigma}_j := (\sigma_j^x, \sigma_j^y, \sigma_j^z)$ and $\vec{h} := (h_x, h_y, h_z)$ encodes the direction and strength of the magnetic field. Clearly, in the limiting case of an infinitely strong magnetic field, the ground state is a product state of all spins polarized in the magnetic field direction. Thus strong magnetic fields lead to a non-topological paramagnetic phase and a quantum phase transition must occur between this phase and the intrinsic topological order of the 3D toric code.

In order to qualitatively investigate the QP-dynamics, we calculate the sub-leading order effects of the magnetic field perturbatively using the framework of perturbative continuous unitary transformations (pCUT) [77, 78] along the same lines as for the 2D toric code in a magnetic field [26, 27, 31]. Note that here we do not aim at a high-order linked-cluster expansion which can be used to pinpoint potential second-order quantum phase transitions, but we want to describe the main effects of the magnetic field and its direction on the dynamics and interactions of QP.

The application of pCUT demands an equidistant spectrum of the unperturbed part of the considered Hamiltonian bounded from below, which is indeed the case for the unperturbed 3D toric code (1). As a consequence, one can introduce an operator $Q$ counting the total number of QP, and rewrite the unperturbed Hamiltonian as

$$H = E_0 + Q, \tag{18}$$

where $E_0$ is the bare ground-state energy, $E_0 = -2N$ for the 3D toric code with $N$ the number of cubes. The perturbing magnetic field term is split into operators $T_n$ which change the number of energy quanta with respect to $Q$ by $n$ where for the 3D toric code $n \in \{0, \pm 2, \pm 4, \pm 6\}$. One can therefore express the 3D toric code in a uniform magnetic field as

$$H_{\text{TCF}} = E_0 + Q + \sum_{j=-3}^{3} \frac{1}{2j} T_{2j}. \tag{19}$$

The pCUT allows to map the Hamiltonian with perturbation written in the form of Eq. (19), order by order in the perturbation $\vec{h}$ exactly, to an effective QP-number-conserving Hamiltonian

$H_{\text{eff}}$ so that $[H_{\text{eff}}, Q] = 0$. Up to second order in the perturbation parameters $h_x$, $h_y$ and $h_z$, the effective Hamiltonian of the 3D toric code reads

$$H_{\text{eff}} = E_0 + Q + T_0 + \sum_{j=1}^{3} [T_{2j}, T_{-2j}]. \qquad (20)$$

Higher orders add more terms to this effective Hamiltonian. Normal-ordering $H_{\text{eff}}$ allows to extract the QP-conserving effective Hamiltonians in the thermodynamic limit in various QP-sectors. Here we have focused on the $0QP$-, $1e$-, $1m$-, $1e$-$1m$-, $2e$- and $2m$-sectors up to the second order in the perturbation. The results are listed in Tab. 2 and will be discussed briefly in the following.

In the vacuum sector of the 3D as well as the 2D toric code only vacuum fluctuations can occur, due to QP-conserving combinations of creation and annihilation processes of second and higher perturbation orders. Their effect is to shift the ground-state energy. When one $e$-QP is present, it modifies these fluctuations, and when $h_z \neq 0$, it can hop to star supersites up to $n$ links apart according to $n$th-order perturbation theory. In the $2e$-sector one can observe that – starting in second order of the perturbation parameter $h_y$ – there exist short-ranged, weakly attractive interactions between $e$-QP due to the following mechanism: When the two $e$-QP neighbor each other, their energy due to vacuum fluctuations is lower than in the case they do not. Still for $h_z \neq 0$, they can lower their energy by delocalizing, which hints at a second-order phase transition at a finite magnetic field strength $h_z$ due to some kind of Bose-Einstein condensation, in the cases when the $e$-QP drive the quantum phase transition rather than the $m$-QP. Both, hopping of $e$-QP due to $h_z \neq 0$ only and attractive interaction between them due to the transverse field $h_y \neq 0$, occur also in the perturbed 2D toric code [26, 27, 31]. We want to qualitatively compare our results for the $2e$-sector of the perturbed 3D toric code in more detail to the 2D toric code in a transverse field $h_y$ which has been investigated in Ref. [27]:

As the unperturbed 2D toric code is symmetric under the exchange $\sigma^x \leftrightarrow \sigma^z$ , its $m$-QP-dynamics due to $h_x \neq 0$ is identical to its $e$-QP-dynamics due to $h_z \neq 0$ as well as the dynamics of $e$- and $m$-QP due to $h_y \neq 0$. For $\vec{h} = (0, h_y, 0)$, single QP cannot hop and are thus static due to selection rules: the parities of the numbers of QP along diagonals and anti-diagonals $(m_x, m_y)$ with sites $(x, y) \in \{m_x b_x + m_y b_y + n \cdot (b_x + b_y)/2 | n \in \mathbb{Z}\}$ and $(x, y) \in \{m_x b_x + m_y b_y + n \cdot (b_x - b_y)/2 | n \in \mathbb{Z}\}$, respectively, are symmetries and thus conserved. The presence of one QP only modifies the vacuum fluctuations. The 2QP-sector of the effective Hamiltonians resulting from pCUT can be further subdivided in the following way: the $1e$-$1m$-sector is not connected to the sector of two $e$-QP and the sector of two $m$-QP, as the perturbation $\sigma^y$ and any product thereof cannot change the parities of the overall numbers of $e$-QP and $m$-QP. In the former sector, the states of the two QP being located on one diagonal or anti-diagonal, as defined above, have lower energies than states where this is not the case. The reason is that when only two (anti-)diagonal parities are odd, one-dimensional correlated hopping of the QP-pair along this (anti-)diagonal direction is possible. This is an example of the phenomenon of dimensional reduction. Furthermore, the closer the QP are, the stronger is the modification of the vacuum fluctuations, the lower is the perturbation order in which correlated hopping appears and thus the lower are the energies of their states. Beside these processes, the sector of two $e$-QP and the sector of two $m$-QP features another process: transmutations of two $e$-QP to two $m$-QP and vice versa, conserving all parities described above. The lowest perturbation order for the transmutation depends on the distance of the QP, too. Both correlated hopping and transmutation imply a short-ranged attractive interaction which leads to the formation of bound states.

In contrast to that, the $2e$-sector of the perturbed 3D toric code is not connected to its $2m$-sector, because such a transmutation is not possible due to the differences in the star and plaquette operators and the lattices of $e$-QP- and $m$-QP-supersites. One-dimensional corre-

Table 2: Effective Hamiltonians of the 3D toric code resulting from pCUT up to second order in the perturbations $h_x$, $h_y$ and $h_z$. $E_0^{(2)}/N$ is the ground-state energy per unit cell. The state $|(e/m);i\rangle$ denotes the state where a $(e/m\text{-})$QP is located at (star/plaquette) supersite $i$ and the label $e/m$ is omitted if it can be inferred from the considered sector. The notation $<i,j>_{(2)}$ means that supersites $i$ and $j$ are one (two) link(s) apart. When an $e$- and an $m$-QP share two common spins, their state is denoted as $|f\rangle$. When two $m$-QP share a common spin $s$, their state is denoted as $|2m;s,1\rangle$ or for the sake of brevity as $|s,1\rangle$, if the context implies that it is a state of two $m$-QP. The complementary configuration of two $m$-QP sharing the same common spin $s$, obtained from $|s,1\rangle$ via the application of $\sigma_s^x$, is denoted by $|s,2\rangle := \sigma_s^x |s,1\rangle$. The factor $p_{ij(l)}$ denotes the number of physical paths between supersites $i$ and $j$, which could depend on whether supersite $l$ is occupied or not. All results beside the ground-state energy of the 0QP-sector are measured with respect to the ground-state energy $E_0^{(2)}$.

| QP-sector | Hamiltonian of pCUT up to second order in $h_x$, $h_y$ and $h_z$ |
|---|---|
| 0QP | $\frac{E_0^{(2)}}{N} = -4 - 3 \cdot (\frac{h_x^2}{4} + \frac{h_y^2}{6} + \frac{h_z^2}{2})$ |
| 1e | $H_{1e}^{(2)} = 1 + 6\frac{h_y^2}{6} - 6\frac{h_y^2}{4} + 6\frac{h_z^2}{2}$ <br> $\quad - h_z \sum_{<i,j>} |i\rangle\langle j| - \frac{h_z^2}{2} p_{ij} \sum_{<i,j>_2} |i\rangle\langle j| + \text{h.c.}$ |
| 1m | $H_{1m}^{(2)} = 1 + 4\frac{h_x^2}{4} - 4\frac{h_x^2}{2} + 4\frac{h_y^2}{6} - 4\frac{h_y^2}{4}$ |
| 2e | $H_{2e}^{(2)} = 2 + \frac{h_z^2}{2} + 11\frac{h_z^2}{2}$ <br> $\quad - h_z \sum_{<i,j>,l\neq i,j} \left( |i,l\rangle\langle j,l| + |l,i\rangle\langle l,j| \right) + \text{h.c.}$ <br> $\quad - \frac{h_z^2}{2} p_{ijl} \sum_{<i,j>_2,l\neq i,j} \left( |i,l\rangle\langle j,l| + |l,i\rangle\langle l,j| \right) + \text{h.c.}$ <br> $\quad + \left( -\frac{h_y^2}{2} - 10\frac{h_y^2}{4} + 11\frac{h_y^2}{6} \right) \cdot \sum_{<i,j>} |i,j\rangle\langle i,j|$ <br> $\quad + \left( -12\frac{h_y^2}{4} + 12\frac{h_y^2}{6} \right) \cdot \left( \mathbb{1} - \sum_{<i,j>} |i,j\rangle\langle i,j| \right)$ |
| 2m | $H_{2m}^{(2)} = 2 - h_x \sum_s |s,1\rangle\langle s,2| + \text{h.c.}$ <br> $\quad + \left( -6\frac{h_x^2}{2} + 7\frac{h_x^2}{4} - \frac{h_y^2}{2} - 6\frac{h_y^2}{4} + 7\frac{h_y^2}{6} \right) \cdot \sum_{s;c=1,2} |s,c\rangle\langle s,c|$ <br> $\quad + \left( -8\frac{h_x^2}{2} + 8\frac{h_x^2}{4} - 8\frac{h_y^2}{4} + 8\frac{h_y^2}{6} \right) \cdot \left( \mathbb{1} - \sum_{s;c=1,2} |s,c\rangle\langle s,c| \right)$ |
| 1e-1m | $H_{1e,1m}^{(2)} = 2 + 4\frac{h_x^2}{4} - 4\frac{h_x^2}{2} + 6\frac{h_z^2}{2}$ <br> $\quad - h_z \sum_{<i,j>} |e;i\rangle\langle e;j| - \frac{h_z^2}{2} p_{ij} \sum_{<i,j>_2} |e;i\rangle\langle e;j| + \text{h.c.}$ <br> $\quad + \left( -2\frac{h_y^2}{2} - 6\frac{h_y^2}{4} + 8\frac{h_y^2}{6} \right) \cdot \sum_f |f\rangle\langle f|$ <br> $\quad + \left( -8\frac{h_y^2}{4} + 8\frac{h_y^2}{6} \right) \cdot \left( \mathbb{1} - \sum_f |f\rangle\langle f| \right)$ |

lated hopping due to $h_y \neq 0$ is not occurring in first- and second-order pCUT, either, since this would create additional $m$-QP and is hence forbidden by QP-number conservation. The exchange $\sigma^x \leftrightarrow \sigma^z$ is not a symmetry of the 3D toric code and, unlike $e$-QP of the perturbed 3D toric code, a single $m$-QP cannot move by *any* perturbation, while the motion of single QP

of the 2D toric code due to $h_y \neq 0$ is forbidden only by a selection rule. For a perturbation of the 3D toric code with non-zero $h_x$ and $h_y$, only the vacuum fluctuations are modified. The reason is that each state with an $m$-QP localized at a different position belongs to a different superselection sector, as discussed for $m$-QP of the 3D toric code above. Such immobility of a QP in a *translationally invariant* system is unusual, as normally disorder causes localization while breaking translational invariance. Two $m$-QP cannot move either, if they do not neighbor each other (share a common spin $s$), but if they do, perturbations by $h_x \neq 0$ can toggle between the complementary configurations of two $m$-QP around the spin $s$. Additionally, such states have a lower energy than two isolated $m$-QP due to the vacuum fluctuations, analogous to the case of two $e$-QP. As a consequence, the lowest-energy states of the $2m$-sector are superpositions of the two complimentary configurations of a pair of $m$-QP localized around a spin $s$. The smallest mobile $m$-QP-configuration is the $4m$-loop, beginning to move in the second order of the perturbation $h_x \neq 0$. This immobility of a single $m$-QP as well as the reduced mobility in all pure $m$-QP sectors in a translationally invariant system is shared by and is a defining property of so-called fracton phases, a topic currently much under investigation and discussion, e.g., see [62, 63, 79]. Therefore we have investigated all $m$-QP-sectors up to the $4m$-sector in the following way: we computed and diagonalized the effective Hamiltonian resulting from second-order pCUT for each sector. Then we have compared the respective lowest energy level of these sectors with each other for perturbation strengths up to the exact phase transition point $\vec{h} = (0, 0, 1/2)$, see Sect. 3, as well as up to the approximative phase transition points according to the variational calculation introduced in Sect. 4 and presented in Sect. 5. It turned out that the resulting energy levels lie higher than the lowest energy levels of the $1m$- and $2m$-sectors in this parameter regime. On this basis these sectors of larger numbers of $m$-QP seem to be irrelevant for the phase transitions; we suspect that this remains true for higher-order pCUT and based our qualitative interpretation of the results in Sect. 5 on it. This suggests that $m$-QP drive a first-order phase transition via some kind of nucleation of a finite density of $m$-QP. The mechanism could be the same as for the first-order phase transition of the 2D toric code in a transverse field, because in both cases single QP are immobile and two neighboring QP can toggle between different configurations, but the 2D toric code in a transverse field additionally features correlated hopping [27].

In the $1e$-$1m$-sector, no new phenomena beside the vacuum fluctuations, the hopping of the $e$-QP, the immobility of the $m$-QP and short-ranged, weakly attractive interactions between $e$-QP and $m$-QP due to $h_y \neq 0$ occur, analogous to the interactions between $e$-QP and to the QP-dynamics of the perturbed 2D toric code [27, 31], but no one-dimensional correlated hopping of neighboring $1e$-$1m$ pairs occurs. The $1e$-$4m$-sector is interesting, because it is the sector with the smallest number of QP such that the exotic mutual braiding statistics featured by the 3D toric code can play a role. To account for the phase of $-1$ resulting from the anticommutation of Pauli matrices applied to the spin in the center of the loop, one can simply change the effective amplitude $t$ for hoppings through the loop to $-t$. We have studied finite systems at $h_x = 0, h_z \neq 0$ using exact diagonalization, but the results showed that the difference in the eigenenergies and -states of the system with and without $4m$-loop and exotic mutual statistics diminishes for increasing system sizes. We expect that in the thermodynamic limit differences could arise only if the density of $4m$-loops is finite, which is not the case for a low-energy state. For $h_x \neq 0$, the $4m$-loop can move, but the hopping resulting from the second and third order of the perturbation is not affected by the mutual statistics; only some hopping processes emerging in fourth and higher orders are modified by it.

Beside this effect of the statistics irrelevant for the phase transitions, it will modify certain effective hopping and vacuum fluctuation amplitudes in sectors of lower QP-numbers in higher-order perturbation theory, as soon as the respective order allows processes like (1) the creation of a $4m$-loop in combination with (2) motion of an $e$-QP in a closed path through the loop back

Table 3: Summary of the physical implications of the results of second-order perturbation theory applied to the 3D toric code in a uniform magnetic field.

| QP-sector | qualitative processes | interpretation |
|---|---|---|
| 0QP | vacuum fluctuations | |
| 1e | hopping, fluctuations | Bose-Einstein condensation → 2$^{\text{nd}}$-order phase transition |
| 1m | immobility, fluctuations | superselection sectors |
| 2e | as for 1e, and short-ranged, weakly attractive interaction | gas of interacting hardcore bosons |
| 2m | superpositions, fluctuations, attractive interaction | bound states, dominate over 1m → 1$^{\text{st}}$ order transition (nucleation) |
| 1e-1m | as for 1e and 1m, and short-ranged, weakly attractive interaction | no bound states for $N \to \infty$ |
| 1e-4m | hopping (of 4m-loop for order ≥ 4), mutual statistics | irrelevant for phase transitions |

to its initial position and (3) annihilation of the loop (minimal order: 6). The above discussion of the physical implications of the results of second-order perturbation theory is summarized in Tab. 3. In the case of the 2D toric code in a uniform magnetic field, it has been found that the regions of the phase diagram with first-order phase transitions and those with second-order phase transitions can roughly be characterized by the criterion whether to the lowest relevant orders in perturbation theory attractive interactions dominate over the kinetic energy (resulting in bound states) or vice versa. This guidance translates to the 3D toric code in the following way: for $h_y \neq 0, h_x = h_z = 0$ there is no kinetic energy and the induced interactions are always attractive; this is true in the whole $h_x$-$h_y$ plane in the regime relevant for the phase transitions and hence we expect first-order phase transitions. For $h_z \neq 0, h_x = h_y = 0$ there is only kinetic energy and thus we expect a second-order phase transition. These limiting cases are separated by the surface for which the elementary energy gaps of the 1e-sector, $\epsilon^{(2)}_{1e,h_z,\Gamma}$, and of the 1m-sector, $E^{(2)}_{1m}$, equal each other, i.e.,[4]

$$\epsilon^{(2)}_{1e,h_z,\Gamma} = 1 - 6h_z - 12h_z^2 \stackrel{!}{=} E^{(2)}_{1m} = 1 - h_x^2 - \frac{h_y^2}{3}. \tag{21}$$

This surface will help us to roughly distinguish regions of first- and second-order quantum phase transitions in the quantum phase diagram for the 3D toric code in a uniform magnetic field discussed in Sect. 5.

Similar to the case of the 2D toric code in a uniform magnetic field, the quantum phase transitions of the perturbed 3D toric code might be driven by the e-QP and 4m-loops, which become dynamical due to the magnetic field, as discussed above. For a general field direction, the investigation of the quantum phase transition requires the application of numerical methods. In [80], Monte Carlo simulations are applied to investigate the phase diagram of the 3D toric code perturbed by *ferromagnetic nearest-neighbor Ising interactions*. The problem posed by the 3D toric code in a uniform magnetic field (17) has not been addressed before in the literature to the best of our knowledge. The quantitative phase diagram of the 2D toric code in a uniform magnetic field, presented comprehensively in [31], has been determined by a combination of various numerical methods, for example quantum Monte-Carlo simulations [23, 29, 30],

---

[4]In the following, the symbol "$\stackrel{!}{=}$" denotes an assumption in contrast to an identity.

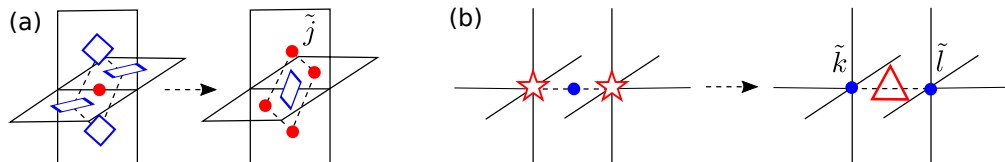

Figure 6: Duality transformations of the perturbed 3D toric code in the cases (a) $\vec{h} = (h_x, 0, 0)$ and (b) $\vec{h} = (0, 0, h_z)$. The left and and right parts of (a) and (b) show the degrees of freedom and interactions before and after the duality transformation, respectively. The red (blue) points depict the action of a Pauli matrix $\sigma_j^x$ or $\tau_{\tilde{j}}^x$ ($\sigma_j^z$ or $\tau_{\tilde{j}}^z$) due to the magnetic field. The squares represent plaquette operators, the stars star operators, and the triangle a nearest-neighbor Ising interaction $\tau_{\tilde{k}}^x \tau_{\tilde{l}}^x$. The dashed lines indicate interactions between the degrees of freedom induced by the magnetic fields in the original toric code picture.

high-order linked-cluster expansions [26,27,31], exact diagonalization [25,27,31,32], tensor network approaches like iPEPS [31] or other variational methods [81]. For the 3D toric code in a generic field, quantum Monte-Carlo simulations are problematic due to the sign problem, exact diagonalizations are limited due to finite cluster sizes, tensor network approaches become challenging in 3D, and linked-cluster expansions are challenging when first- and second-order quantum phase transitions are present in the quantum phase diagram. As a consequence, we combine exact dualities and variational approaches to tackle this problem.

## 3  Exact duality relations

Similarly to the 2D toric code in a uniform magnetic field, it is possible to find exact duality relations in the 3D case for specific field directions. This allows to pinpoint the location and the order of the quantum phase transition in some cases exactly. In addition, one can benchmark the quality of our variational approach discussed in Sect. 4.

**Duality transformation for $\vec{h} = (h_x, 0, 0)$.** – For this magnetic field direction, the star operators $A_s$ commute with the Hamiltonian and therefore label different Hilbert space sectors. Thus to investigate the physics at low energies, one can set their eigenvalues to $a_s = +1 \ \forall s$. The Hamiltonian of this low-energy sector reads

$$H^x(h_x; \sigma) := -\frac{N}{2} - \frac{1}{2} \sum_p B_p - h_x \sum_j \sigma_j^x. \tag{22}$$

In the following we use $\lambda_x := 2h_x$ and denote the center of the plaquettes $p$ to be the sites $\tilde{j}$ of the dual lattice. The original and dual lattice are identical, but shifted by a constant vector. Notice that the application of a Pauli matrix $\sigma_j^x$ flips the eigenvalues of the four plaquette operators $B_p$, $p = 1, 2, 3, 4$ surrounding any spin $j$ (see Fig. 6 (a)). Hence we define new variables

$$\tau_{\tilde{j}}^x := B_p \qquad \Rightarrow \qquad \sigma_j^x = \prod_{\tilde{j}=1}^4 \tau_{\tilde{j}}^z =: B_{\tilde{p}}, \tag{23}$$

which obey the same (anti-)commutator relations as the operators $\sigma_j^x$ and $B_p$. The dual Hamil-

tonian in the new variables turns out to be the same as in the original variables:

$$H_{\text{dual}}^x(\lambda_x;\sigma) = -\frac{N}{2} - \frac{1}{2}\sum_{\tilde{j}}\tau_{\tilde{j}}^x - \frac{\lambda_x}{2}\sum_{\tilde{p}}B_{\tilde{p}} = \lambda_x H^x(\lambda_x^{-1};\tau), \tag{24}$$

$$E(\lambda_x) = \lambda_x E(\lambda_x^{-1}),$$

i.e., the Hamiltonian is *self-dual*. If there exists only a unique phase transition point – which is physically reasonable – of the Hamiltonian $H^x(\lambda_x;\sigma)$ at $\lambda_x^c = 2h_x^c$, $H_{\text{dual}}^x(\lambda_x^{-1};\tau)$ must have one phase transition point at $(\lambda_x^c)^{-1}$. The uniqueness can only hold for $\lambda_x^c = 1$, which implies

$$h_x^c = \frac{1}{2}. \tag{25}$$

Furthermore, it can be shown that the effective Hamiltonian $H^x$ is equivalent to the self-dual four-dimensional version of Wegner's lattice gauge theory [82]. This model exhibits a single first-order phase transition [83, 84] and therefore the phase transition of the 3D toric code perturbed by $h_x$ is known to be of first order.

**Duality transformation for $\vec{h} = (0, 0, h_z)$.** – In this case, the plaquette operators $B_p$ commute with the Hamiltonian. Analogously to the case before, the physics at low energies takes place in the sector where $b_p = +1 \; \forall p$ and the Hamiltonian reduces to

$$H^z(h_z;\sigma) := -\frac{3N}{2} - \frac{1}{2}\sum_s A_s - h_z\sum_j \sigma_j^z. \tag{26}$$

We define new variables

$$\tau_{\tilde{j}}^z := A_s \quad \Rightarrow \quad \sigma_j^z = \tau_{\tilde{k}}^x\tau_{\tilde{l}}^x \quad \Rightarrow \quad \tau_{\tilde{k}}^x = \prod_{n\in\mathbb{N}^0}\sigma_{\tilde{k}+(2n+1)b_\beta/2}^z, \qquad \lambda_z := 2h_z, \tag{27}$$

where the indices $\tilde{j}, \tilde{k}, \tilde{l}$ label centers of stars as illustrated in Fig. 6 (b). In contrast to the case of a non-zero $h_x$, the original and the dual lattice, which is simple cubic, are not identical. The subscript $\tilde{j} + (2n+1)b_\beta/2 \in \Lambda$, with $b_\beta$ as in Eq. (9) and $\beta \in \{x, y, z\}$, denotes spin sites of the original lattice $\Lambda$ forming a *string* which starts at site $\tilde{j} + b_\beta/2$ and goes to infinity in the freely chosen $\beta$-direction. This amounts to a non-local (topological) transformation. The new variables satisfy the same (anti-)commutation relations as the original operators $\sigma_j^z$ and $A_s$. Thus the dual Hamiltonian [85] is given by

$$H_{\text{dual}}^z(\lambda_z;\sigma) = -\frac{3N}{2} - \sum_{\tilde{j}}\tau_{\tilde{j}}^z - \lambda_z\sum_{<\tilde{k},\tilde{l}>}\tau_{\tilde{k}}^x\tau_{\tilde{l}}^x, \tag{28}$$

which describes the ferromagnetic 3D transverse-field Ising model (3D TFIM). The 3D TFIM is not exactly solvable, but various publications, e.g., [86, 87], determined the zero-temperature quantum critical point numerically to be

$$\lambda_z^c \approx 0.194 \quad \Leftrightarrow \quad h_z^c \approx 0.097 \tag{29}$$

via series expansion techniques. These results were confirmed by other methods, like (Quantum) Monte Carlo techniques in [88, 89]. The quantum phase transition between the Ising-ordered low-field and the paramagnetic high-field phase is of second order. It belongs to the $(3 + 1)$D-Ising universality class and has mean-field critical exponents. The corresponding quantum criticality in the dual picture, i.e., for the 3D toric code in a uniform magnetic field $h_z$, is then $(3 + 1)$D-Ising* [90].

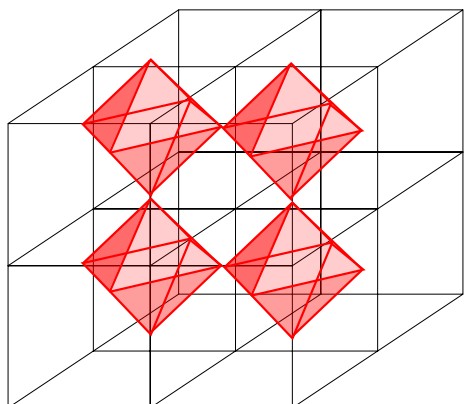 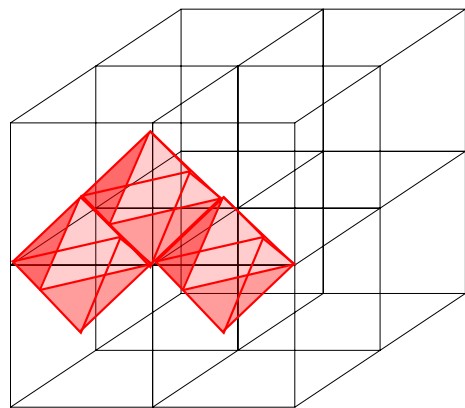

Figure 7: Different spatial configurations of star interactions (tetrahedra) in the case $\vec{h} = (0, 0, h_z)$ (26) (left) and $\vec{h} = (0, h_y, 0)$ (31) (right). In the left case tetrahedra can share spins at corners; in the right case tetrahedra can share spins at common corners or edges.

**Duality transformation for $\vec{h} = (0, h_y, 0)$.** – This field configuration is also called the toric code in a transverse field. Here neither the star nor the plaquette operators commute with the Hamiltonian. Hence one cannot simplify the Hamiltonian as in the two previous cases. Nevertheless, a duality transformation determined by the variables

$$
\tau^z_{\tilde{j}_s} := A_s, \qquad \tau^z_{\tilde{j}_p} := B_p \qquad \Rightarrow \qquad \sigma^y_j = \prod_{\tilde{j}_s : j \in \tilde{j}_s}^{2} \tau^x_{\tilde{j}_s} \prod_{\tilde{j}_p : j \in \tilde{j}_p}^{4} \tau^x_{\tilde{j}_p} =: A_{\tilde{s}}, \qquad \lambda_y := 2h_y, \quad (30)
$$

allows to obtain the dual Hamiltonian[5]

$$
H^y(\lambda_y; \sigma) = -\frac{1}{2} \sum_{\tilde{j}} \tau^z_{\tilde{j}} - \frac{\lambda_y}{2} \sum_{\tilde{s}} A_{\tilde{s}}. \tag{31}
$$

Hamiltonian (31) is a generalization of the 2D Xu-Moore model [91, 92] to 3D. To the best of our knowledge, no information on the location and the order of the phase transition are known and therefore the exact duality relation does not provide any insights into the breakdown of the topological phase in this case. However, a first-order phase transition might be expected, as also deduced variationally in Sect. 4.

Still, we can check whether the 3D toric code in a transverse field is self-dual, since this implies that the coefficients of a perturbative expansion of the ground-state energy around $\lambda_y^{-1} = 0$ and $\lambda_y = 0$ match each other order-by-order, see Eq. (52) in App. A. One finds

$$
\text{for } \lambda_y > \lambda_y^c: \quad \frac{E(\lambda_y)}{N} = -\frac{3}{2} - \frac{1}{6}\left(-\frac{1}{2\lambda_y}\right)^2 - \frac{3}{4}\left(-\frac{1}{2\lambda_y}\right)^2 + \mathcal{O}\left(\frac{1}{\lambda_y^3}\right) = -\frac{3}{2} - \frac{11}{48}\frac{1}{\lambda_y^2} + \mathcal{O}\left(\frac{1}{\lambda_y^3}\right),
$$

$$
\text{for } \lambda_y < \lambda_y^c: \quad \frac{E(\lambda_y)}{N} = -2 - \frac{h_y^2}{2}\mathcal{O}(\lambda_y^3) = -2 - 2\lambda_y^2 + \mathcal{O}(\lambda_y^3),
$$

$$
\Rightarrow \qquad E(\lambda_y) \neq \lambda E(\lambda_y^{-1}). \tag{32}
$$

---

[5](subscripts $s$ and $p$ will be dropped in the following for the sake of brevity)

So self-duality is absent in this field direction. In the expression for $\lambda_y > \lambda_y^c$, the first term is the energy of the unperturbed Hamiltonian, first-order corrections are absent and in second order, vacuum fluctuations due to the star interactions (second term) and plaquette interactions (third term) occur. The physical reason for the model being not self-dual is that the dynamics and fluctuations of $e$-QP and $m$-QP due to the magnetic field is different from the magnons' dynamics due to the star and plaquette operators, because $e$-QP and $m$-QP and magnons hop on different lattices of supersites.

Altogether, the nature of the quantum phase transition between the 3D topologically-ordered and the polarized phase depends on the field direction. For $\vec{h} = (h_x, 0, 0)$, a first-order phase transition takes place exactly at $h_x^c = 0.5$ due to self-duality. In contrast, the transition is of second order in the (3+1)D-Ising* universality class for $\vec{h} = (0, 0, h_z)$ with $h_z^c \approx 0.097$ [86,87].

# 4 Variational approaches

We use a variational approach to determine the quantum phase diagram of the perturbed 3D toric code as presented in Sect. 5. Inspired by Ref. [81], the following ansatz for the ground-state wave function of the 3D toric code in a uniform magnetic field with variational parameters $\alpha, \beta$ is chosen:

$$|\alpha, \beta\rangle := \mathcal{N}(\alpha, \beta) \prod_s (\mathbb{1} + \alpha A_s) \prod_p (\mathbb{1} + \beta B_p) |\vec{h}\vec{h}\ldots\vec{h}\rangle \,, \qquad \alpha, \beta \in [0, 1] \,, \qquad (33)$$

where $\mathcal{N}(\alpha, \beta) \equiv \mathcal{N}$ is a normalization constant. The ket $|\vec{h}\vec{h}\ldots\vec{h}\rangle$ denotes the state of all spins pointing in the direction of the magnetic field. For the sake of brevity, the notation $|\vec{h}\rangle \equiv |\vec{h}\vec{h}\ldots\vec{h}\rangle$, $|\alpha\rangle \equiv |\alpha, \beta = 1\rangle$, $|\beta\rangle \equiv |\alpha = 1, \beta\rangle$ and $\mathcal{N}(\alpha) \equiv \mathcal{N}(\alpha, \beta = 1)$, $\mathcal{N}(\beta) \equiv \mathcal{N}(\alpha = 1, \beta)$ is used in the remainder of this post. Most importantly, the two limiting cases $\alpha = \beta = 1$ and $\alpha = \beta = 0$ are exactly equal to the toric code ground state (4) for $|\vec{h}| = 0$ and to the polarized ground state $|\vec{h}\rangle$ for $|\vec{h}| = \infty$, respectively. For $\alpha = \beta = 1$, the normalization is known to be $\mathcal{N}(1, 1) = 2^{-4N}$, where $N$ is the number of unit cells, and $\mathcal{N}(0, 0) = 1$.

Transferring the ideas of [93] to the 3D toric code in a uniform magnetic field, ansatz (33) can be reformulated: Let $\mathcal{P}^m$ label closed membranes of spins in the state $\sigma^x |\vec{h}\rangle$, i.e., generated by products of $A_s$, and let $\mathcal{L}^e$ label closed loops of spins in the state $\sigma^z |\vec{h}\rangle$ as shown in Fig. 2 of SubSect. 2.1. Then

$$|\alpha, \beta\rangle = \mathcal{N}\Big(\mathbb{1} + \alpha \sum_{s_1} A_{s_1} + \beta \sum_{p_1} B_{p_1} + \alpha^2 \sum_{s_1, s_2, s_1 \neq s_2} A_{s_1} A_{s_2} + \alpha\beta \sum_{s_1, p_1} A_{s_1} B_{p_1} + \ldots\Big) |\vec{h}\rangle$$
$$= \mathcal{N} \sum_{\mathcal{P}^m, \mathcal{L}^e \text{ closed}} (\alpha^{1/6})^{A(\mathcal{P}^m)} (\beta^{1/4})^{L(\mathcal{L}^e)} |\mathcal{P}^m, \mathcal{L}^e\rangle \,, \qquad (34)$$

where $A(\mathcal{P}^m)$ $(L(\mathcal{L}^e))$ is a function which represents the area (length) of membranes $\mathcal{P}^m$ (loops $\mathcal{L}^e$). So one can think of $\alpha^{1/6}$ $(\beta^{1/4})$ as the inverse of some kind of surface (string) tension competing for example with some kind of kinetic energy. Consequently, the amplitudes of the states in the superposition forming the ground state $|\alpha, \beta\rangle$ are weighted according to the area (length) of their membranes (loops).

In the specific single-field case $\vec{h} = (0, 0, h_z)$ with $\beta = 1$, the variational ansatz (33) turns out to be of mean-field character in the sense that for any set of $n$ stars $\mathcal{S}_n$

$$\langle \prod_{s \in \mathcal{S}_n} A_s \rangle_\alpha := \langle \alpha| \prod_{s \in \mathcal{S}_n} A_s |\alpha\rangle = \eta^n = \big(\langle A_s \rangle_\alpha\big)^n \,, \qquad (35)$$

where $\eta := 2\alpha/(1 + \alpha^2)$. In contrast to the perturbed 2D toric code [81], this is not true for $\vec{h} = (h_x, 0, 0)$ and other field configurations, since for a set of $n$ plaquettes $\mathcal{P}_n$, irrespective of being linked or not, one has

$$\langle \prod_{p \in \mathcal{P}_n} B_p \rangle_\beta := \langle \beta | \prod_{p \in \mathcal{P}_n} B_p |\beta\rangle = \mathcal{N}^2(\beta)(1+\beta^2)^{3N} \langle \Rightarrow| \prod_{p \in \mathcal{P}_n} (\zeta \mathbb{1} + B_p) \prod_{p' \notin \mathcal{P}_n} (\mathbb{1} + \zeta B_{p'})| \Rightarrow\rangle \, , \quad (36)$$

with $N$ the number of unit cells; defining $\zeta := 2\beta/(1 + \beta^2)$ and $|\Rightarrow\rangle := |\rightarrow\rightarrow \cdots \rightarrow\rangle$. This is not necessarily equal to $(\langle B_p\rangle_\beta)^n = \zeta^n$, for instance when the set of plaquettes $\mathcal{P}_n$ forms an elementary cube or any other closed membrane such that the product of plaquette operators is the identity. General configurations $\vec{h} = (h_x, h_y, h_z)$ with $\alpha, \beta \neq 1$ also lead to certain products of star and plaquette operators proportional to the identity. Therefore the variational ansatz (33) goes beyond mean-field theory.

In practice, one computes the variational ground-state energy per spin $e(\alpha, \beta) := \langle H\rangle_{\alpha,\beta}/(rN)$ for this variational ansatz with $r = 3$ the number of spins per unit cell. Then one minimizes the energy with respect to the variational parameters $\alpha$ and $\beta$ in order to identify different phases. In the following we discuss first the specific single-field cases in $h_x$-, $h_y$-, and $h_z$-direction, where we were able to obtain the analytical solution of the variational calculation. Afterwards, an approximative approach to the general-field case is presented.

## 4.1 Single-field cases

### 4.1.1 $h_z$-field

In this field direction $\vec{h} = (0, 0, h_z)$, ansatz (33) simplifies to

$$|\alpha\rangle = \mathcal{N}(\alpha) \prod_s (\mathbb{1} + \alpha A_s) |\Uparrow\rangle \, , \quad (37)$$

as the plaquette operators $B_p$ commute with the Hamiltonian (26). Thus $\beta = 1$ ensures that the product wave function $|\Uparrow\rangle := |\uparrow\uparrow \ldots \uparrow\rangle$ is projected onto the low-energy Hilbert space sector with $b_p = +1 \ \forall p$. The variational energy per spin, as derived in App. B, is minimal at

$$\eta = \alpha = 1 \quad \text{for } h_z < \frac{1}{12}, \qquad \eta = \frac{1}{12h_z} \quad \text{for } h_z \geq \frac{1}{12}, \quad (38)$$

with the limiting case $\alpha = 0$ for the fully polarized phase at $h_z \to \infty$. The minimal energies for these two cases are

$$e(\eta = 1) = -\frac{2}{3}, \qquad e(\eta = \frac{1}{12h_z}) = -\frac{1}{144h_z} - \frac{1}{2} - h_z \, , \quad (39)$$

which match at $h_z^c = 1/12$ without a kink. This indicates a second-order quantum phase transition at the point $h_z^c = \frac{1}{12}$, which is 14% off the preciser point $h_z^c \approx 0.097$ for the second-order phase transition of the 3D TFIM [86], as discussed in the last Sect. 3.

### 4.1.2 $h_x$-field

For $\vec{h} = (h_x, 0, 0)$ one can simplify ansatz (33) to

$$|\beta\rangle = \mathcal{N}(\beta) \prod_p (1 + \beta B_p) |\Rightarrow\rangle \, , \quad (40)$$

as the star operators $A_s$ commute with the Hamiltonian (22) and thus $\alpha = 1$ ensures that the product wave function $|\Rightarrow\rangle$ is projected onto the low-energy Hilbert space sector $a_s = +1 \ \forall s$. The normalization constant equals

$$
\frac{\langle\beta|\beta\rangle}{\mathcal{N}^2(\beta)(1+\beta^2)^{3N}} = \langle\Rightarrow|\prod_p(\mathbb{1}+\zeta B_p)|\Rightarrow\rangle =
$$

$$
= \langle\Rightarrow|\mathbb{1} + \zeta^6\sum_{\mathcal{C}_6}\prod_{p\in\mathcal{C}_6}B_p + \zeta^{10}\sum_{\mathcal{C}_{10}}\prod_{p\in\mathcal{C}_{10}}B_p + \zeta^{12}\sum_{\mathcal{C}_6^2}\prod_{p\in\mathcal{C}_6^2}B_p + \zeta^{14}\sum_{\mathcal{C}_{14}}\prod_{p\in\mathcal{C}_{14}}B_p + \ldots|\Rightarrow\rangle = \quad (41)
$$

$$
= 1 + N\zeta^6 + 6N\zeta^{10} + N(N-7)\zeta^{12} + 10\cdot6N\zeta^{14} + \ldots \overset{!}{=} 1\,,
$$

where $\mathcal{C}_n$ is the (product of) closed cube constraints in Tab. (1) with $n$ faces and all other terms not proportional to the identity cancel due to orthogonality of the different states. The constraints can be thought of as constituted of elementary cubes $\mathcal{C}_6$ of plaquette operators, illustrated in Fig. (3) of SubSect. 2.1. Consequently, the sum over all cubes $\mathcal{C}_6$ contains $N$ terms, where $N$ is the number of unit cells. The notation $\mathcal{C}_n^m$ denotes $m$ unconnected cubes, each with $n$ faces. For the coefficients of the terms of higher orders in $\zeta$, one has to count the number of positions to place the respective combinations of cube constraints in the system. The variational energy per spin can be obtained by calculating the expectation values of star, plaquette and spin operators. For a star operator it is:

$$
a_s := \langle\beta|A_s|\beta\rangle = 1\,, \tag{42}
$$

and the expectation value of a plaquette operator yields:

$$
b_p(\beta) := \langle\beta|B_p|\beta\rangle = \mathcal{N}^2(\beta)(1+\beta^2)^{3N}\langle\Rightarrow|(\zeta\mathbb{1}+B_p)\prod_{p',p'\neq p}(\mathbb{1}+\zeta B_{p'})|\Rightarrow\rangle =
$$

$$
\overset{(41)}{=} \frac{\zeta(1+(N-2)\zeta^6+(6N-10)\zeta^{10}+\ldots)+(1/\zeta)(2\zeta^6+10\zeta^{10}+\ldots)}{1+N\zeta^6+6N\zeta^{10}+N(N-7)\zeta^{12}+10\cdot6N\zeta^{14}+\ldots} = \quad (43)
$$

$$
= \zeta + \frac{\zeta(-2\zeta^6-10\zeta^{10}+\ldots)+(1/\zeta)(2\zeta^6+10\zeta^{10}+\ldots)}{1+N\zeta^6+6N\zeta^{10}+N(N-7)\zeta^{12}+10\cdot6N\zeta^{14}+\ldots}\,.
$$

Fig. 8 (a) illustrates why in the second line of the formula above in the first bracket of the numerator the second term equals $(N-2)\zeta^6$: two out of $N$ cubes are not contained in the sum over cubes $\mathcal{C}_6$ due to the missing plaquette in the product $\prod_{p',p'\neq p}(\mathbb{1}+\zeta B_{p'})$, indicated in the illustration by the large green crosses. The other parts (b) to (e) illustrate the higher-order coefficients. The first term in the second bracket is in turn explained by Fig. 9 (a), as the elementary cube must contain plaquette $p$; otherwise the product of 6 plaquettes does not equal the identity. The contribution to the variational energy per spin in the thermodynamic limit is

$$
\frac{\sum_p\langle\beta|B_p|\beta\rangle}{3N} = b_p(\beta) \overset{N\to\infty}{\longrightarrow} \zeta\,, \tag{44}
$$

and the second term in the result of (43) vanishes, as the orders of $N$ in the terms of the denominator dominate over the respective terms of the numerator. The expectation value for the magnetic field term ($\sigma_j^x$) can be computed similarly:

$$
s_j(\beta) := \langle\beta|\sigma_j^x|\beta\rangle = \mathcal{N}^2(\beta)(1+\beta^2)^{3N-4}(1-\beta^2)^4\langle\Rightarrow|\sigma_j^x\prod_{p,p\neq p_1,\ldots,p_4}(\mathbb{1}+\zeta B_p)|\Rightarrow\rangle =
$$

$$
\overset{(41)}{=} (1-\zeta^2)^2\frac{1+(N-4)\zeta^6+(6N-20)\zeta^{10}+\ldots}{1+N\zeta^6+6N\zeta^{10}+N(N-7)\zeta^{12}+10\cdot6N\zeta^{14}+\ldots} = \quad (45)
$$

$$
= (1-\zeta^2)^2 + (1-\zeta^2)^2\frac{-4\zeta^6-20\zeta^{10}+\ldots}{1+N\zeta^6+6N\zeta^{10}+N(N-7)\zeta^{12}+10\cdot6N\zeta^{14}+\ldots}
$$

$$
\overset{N\to\infty}{\longrightarrow} (1-\zeta^2)^2\,,
$$

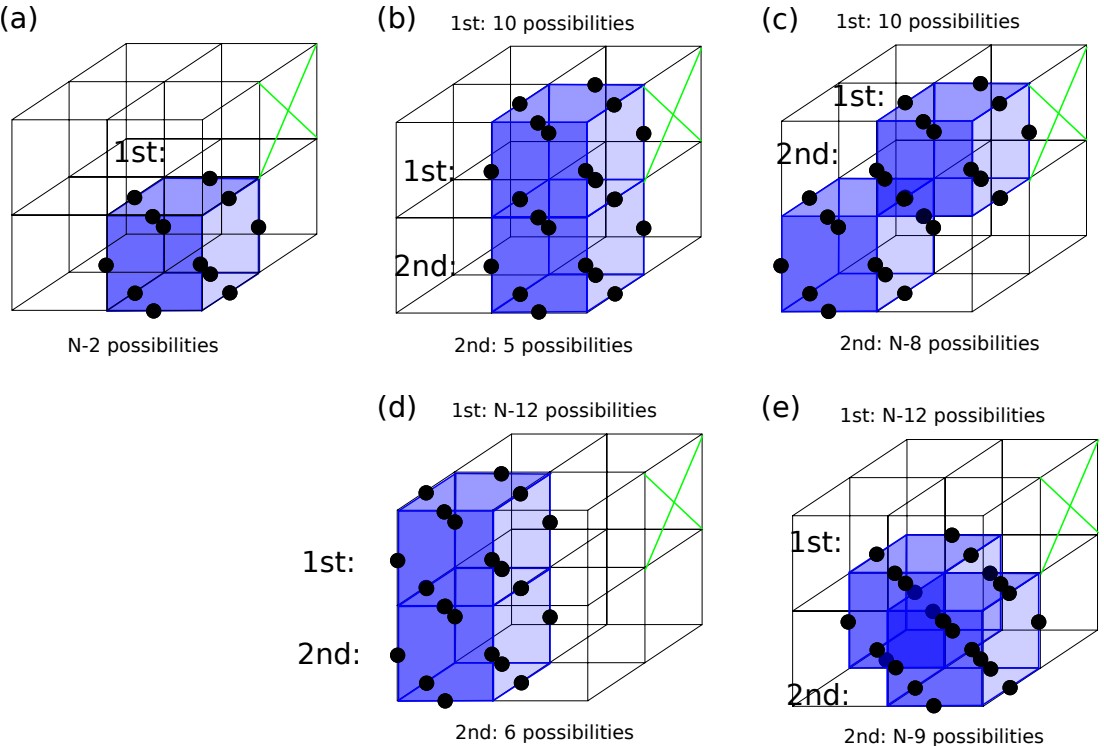

Figure 8: Number of possible positions of products of cube constraints (blue) as discussed in the main text. Here the positions are counted by subsequently placing the "1st" and then the "2nd" cube. The large green cross indicates which face cannot be used to form cube constraints.

where the reasoning and illustrations are similar to before, like Fig. 10 (a) illustrates the second term $(N-4)\zeta^6$ in the numerator. The variational energy per spin in the thermodynamic limit is given by

$$e(\beta) = -\left(\frac{1}{6} + \frac{2\beta}{2(1+\beta^2)}\right) - h_x \left(\frac{1-\beta^2}{1+\beta^2}\right)^4. \tag{46}$$

The minimum of the variational energy $e(\beta)$ for a given magnetic field strength $h_x$ was determined numerically using the function `roots` of NumPy. As result we find a first-order phase transition at $h_x \approx 0.422$ with a jump in the variational parameter $\beta$ (upper part of plot in Fig. 11) and an energy level crossing (lower part) between the solution for the topological phase (green) and for the paramagnetic phase (blue). The result is approximately 15.6% off the self-dual point $h_x = 0.5$ found in the last Sect. 3.

### 4.1.3 $h_y$-field

As the star and plaquette operators commute with respect to each other for $\vec{h} = (0, h_y, 0)$, the calculations for this case are simply a combination of those for the two cases before. The resulting variational energy is given by

$$e(\alpha, \beta) = -\left(\frac{2\alpha}{3 \cdot 2(1+\alpha^2)} + \frac{2\beta}{2(1+\beta^2)}\right) - h_y \left(\frac{1-\alpha^2}{1+\alpha^2}\right)^2 \left(\frac{1-\beta^2}{1+\beta^2}\right)^4. \tag{47}$$

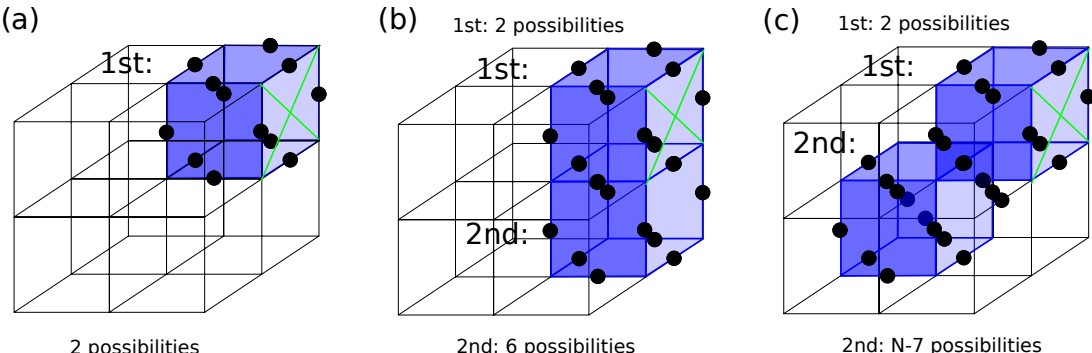

Figure 9: Number of possible positions of products of cube constraints (blue) as discussed in the main text. The labelling is like in Fig. 8, except that the large green cross indicates which face need to be included in the cube constraints.

Its minimization yields a result similar to Fig. 11 for the case $\vec{h} = (h_x, 0, 0)$: a level crossing and thus a first-order phase transition at $h_y \approx 0.615$.

## 4.2 General case - perturbative variational calculations

For the calculations in the general case, it is convenient to work in the basis $\{|\vec{h}\rangle, |-\vec{h}\rangle\}$. In order to do so, the transformations of Pauli matrices and their eigenstates from the commonly used basis $\{|\uparrow\rangle, |\downarrow\rangle\}$ to the basis $\{|\vec{h}\rangle, |-\vec{h}\rangle\}$ are needed. These transformations are conveniently parametrized by the spherical coordinates $\vartheta, \varphi$ of the Bloch sphere; they are displayed in App. C. The result is that in general all Pauli matrices $\sigma^x, \sigma^y, \sigma^z$ in the original basis contain a non-zero component proportional to the Pauli matrix $\sigma_{\vec{h}}^z$ in the rotated basis. This means that not only plaquette operator products which yield the identity, but also other operators contribute to the expectation values, i.e.,

$$\langle \vec{\mathbb{h}} | \sigma_i^\alpha | \vec{\mathbb{h}} \rangle \neq 0, \quad \langle \vec{\mathbb{h}} | A_s | \vec{\mathbb{h}} \rangle \neq 0, \quad \langle \vec{\mathbb{h}} | B_p | \vec{\mathbb{h}} \rangle \neq 0 \qquad \forall i, s, p; \ \alpha \in \{x, y, z\}.$$

This renders the calculation of the variational energy as above intractable, because one has to consider many different configurations contributing differently. Their number increases rapidly with the order of the variational parameters $\eta$ and $\zeta$, as illustrated in Fig. 13 of App. D: in first order only configurations marked with "1" are relevant; in second order already all displayed ones. Thus only the terms to lowest orders in $\eta$ and $\zeta$ can be calculated by hand.

But this obstacle can be turned into a strategy: the quantum phase transition in some magnetic field directions occurs at a discontinuous jump of the variational parameters $\eta, \zeta$ from small values close to 0 to large values close or equal to 1. Then it is reasonable to assume that the lowest-order terms – which can be calculated by hand – approximate the variational energy well for the paramagnetic phase and that this approximation of the ground state energy can be compared to the exact ground state energy $e_{\text{top}} = -2/3$ of the unperturbed toric code. The calculations in SubsubSect. 4.1.2 and 4.1.3 revealed that for example the first-order phase transitions in $(h_x, 0, 0)$- and $(0, h_y, 0)$-directions are of this nature, respectively; see Fig. 11. This amounts to a certain kind of expansion of the variational energy around the high-field

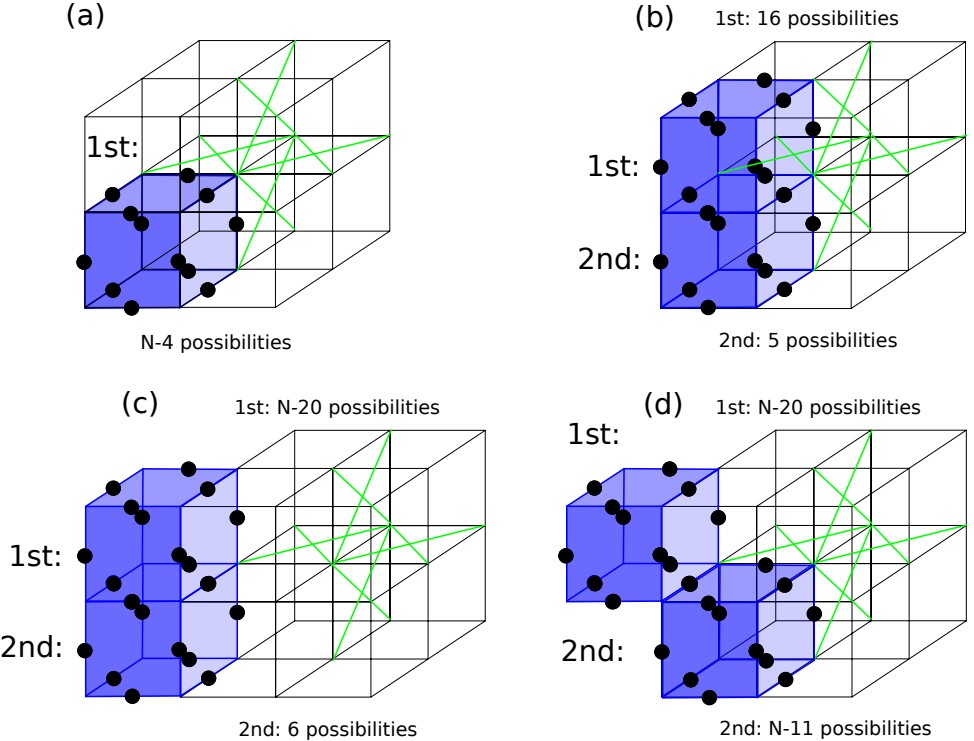

Figure 10: Number of possible positions of products of cube constraints (blue) as discussed in the main text. The labelling is like in Fig. 8.

limit and of the normalization, which is needed to calculate the former:

$$1 \stackrel{!}{=} \langle \alpha, \beta | \alpha, \beta \rangle \approx \mathcal{N}^2(\alpha, \beta)(1+\alpha^2)^N(1+\beta^2)^{3N} \langle \vec{\mathbb{h}} | 1 + \eta \sum_s A_s + \zeta \sum_p B_p + \dots | \vec{\mathbb{h}} \rangle,$$

$$\frac{\langle \alpha, \beta | A_s | \alpha, \beta \rangle}{\mathcal{N}^2(\alpha, \beta)} \approx (1+\alpha^2)^N(1+\beta^2)^{3N} \langle \vec{\mathbb{h}} | (\eta 1 + A_s)(1 + \eta \sum_{s', s' \neq s} A_{s'} + \zeta \sum_p B_p + \dots) | \vec{\mathbb{h}} \rangle,$$

$$\frac{\langle \alpha, \beta | \sigma_j^y | \alpha, \beta \rangle}{\mathcal{N}^2(\alpha, \beta)} \approx \frac{(1-\alpha^2)^2}{(1+\alpha^2)^{2-N}} \frac{(1-\beta^2)^4}{(1+\beta^2)^{4-3N}} \langle \vec{\mathbb{h}} | \sigma_j^y (1 + \eta \sum_{s \neq s_1, s_2} A_s + \zeta \sum_{p \neq p_1, \dots, p_4} B_p + \dots) | \vec{\mathbb{h}} \rangle.$$

All other terms needed for the variational energy have to be treated in the same fashion. App. D summarizes the results for these expansions of the normalization constant and the expectation values $\langle A_s \rangle$, $\langle B_p \rangle$, $\langle \sigma_i^x \rangle$, $\langle \sigma_i^y \rangle$ and $\langle \sigma_i^z \rangle$ up to first order as well as to second order in $\eta$ and $\zeta$. Another advantage of this approach is that one can compute the expansion of the variational energy of the toric code for unspecified lattice coordination numbers and adapt the result afterwards to the specific lattice and dimension. Putting all pieces together yields an expression for the variational energy depending on the lattice coordination numbers, the magnetic field direction $(\vartheta, \varphi)$, the number of unit cells $N$, the magnetic field strength $h \equiv |\vec{h}|$ and the variational parameters $\eta$ and $\zeta$.

The next step is to perform the thermodynamic limit $N \to \infty$. This needs additional care for the single-field cases, as discussed in App. D, too. The result for the variational energy in

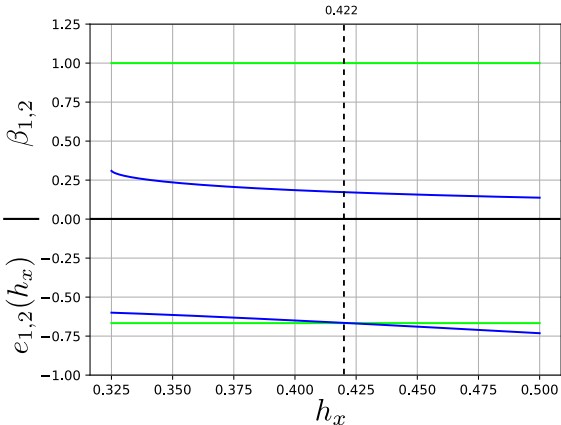

Figure 11: First-order phase transition between the topological phase and the paramagnetic phase in the special case $\vec{h} = (h_x, 0, 0)$. The upper part of the plot shows – in green for the topological phase and in blue for the paramagnetic phase – the different values of the variational parameter minimizing the variational energy for a given $h_x$. The lower part shows in the same color coding the two lowest local minima $e_1(h_x), e_2(h_x)$ of the variational energy. Below $h_x \approx 0.325$ only the solution corresponding to the topological phase exists. The dashed line emphasizes the phase transition at $h_x \approx 0.422$.

the thermodynamic limit up to second order in $\eta$ and $\zeta$ is

$$
\begin{aligned}
e(\eta, \zeta; \vec{h}) =& \left[ -\frac{r_s}{2} \cdot \left( x^{2n_s} \eta + r x^{n_s} z^{n_p} \zeta \right) - \frac{r_p}{2} \cdot \left( x^{n_s} z^{n_p} \eta + r z^{2n_p} \zeta \right) \right. \\
& - (1-\zeta^2)^2 \, hx \cdot \left( x^{n_s+1} \eta + r x z^{n_p} \zeta \right) - \frac{r_s}{r}(1-\eta^2)(1-\zeta^2)^2 \, hy \cdot \left( x^{n_s} y \eta + r y z^{n_p} \zeta \right) \\
& \left. - (1-\eta^2) \, hz \cdot \left( x^{n_s} \eta + r z^{n_p+1} \zeta \right) \right] / \left( x^{n_s} \eta + r z^{n_p} \zeta \right) \\
=& \; -r_s x^{n_s} - r_p z^{n_p} - (1-\zeta^2)^2 \, hx^2 - (1-\eta^2)(1-\zeta^2)^2 \, hy^2 - (1-\eta^2) \, hz^2,
\end{aligned}
$$

(48)

where $r$ denotes the ratio of the number of stars to the number of unit cells, $r_s$ is that of the number of stars to the number of spins, $r_p$ is the number of plaquettes divided by the number of spins, $n_s$ is the number of spins in (stars neighboring) a star and $n_p$ is the number of spins in a plaquette (for the 3D toric code $r = 3$, $r_s = 3$, $r_p = 1$, $n_s = 6$ and $n_p = 4$). We introduced the abbreviations $x := \sin(\vartheta)\cos(\varphi)$, $y := \cos(\vartheta)\sin(\varphi)$ and $z := \cos(\vartheta)$. In the single-field cases the same procedure yields

$$
\begin{aligned}
\text{for } (h_x, 0, 0): \quad & e(\eta, \zeta; h_x) = -\frac{1}{2r} - \frac{1}{2}\zeta - \frac{r_s}{r}(1-\zeta^2)^2 \, h, \\
\text{for } (0, h_y, 0): \quad & e(\eta, \zeta; h_y) = -\frac{1}{2r}\eta - \frac{1}{2}\zeta - \frac{r_s}{r}(1-\eta^2)(1-\zeta^2)^2 \, h, \\
\text{for } (0, 0, h_z): \quad & e(\eta, \zeta; h_z) = -\frac{1}{2r}\eta - \frac{1}{2} - \frac{r_s}{r}(1-\eta^2) \, h,
\end{aligned}
$$

(49)

which are identical to the variational energies of the full variational ansatz in the special cases discussed in SubSect. 4.1. For a general field direction, the expansion up to second order yields results all identical to that of the first order.

In order to obtain the approximative phase diagram for the general case, i.e., to find for all directions of the magnetic field its critical strength where the phase transition occurs, the following steps were executed by a Mathematica script (which rasterizes all magnetic field directions in $90 \times 90$ points):

1. Evaluate $e(\eta, \zeta; \vec{h})$ in Eq. (48) and (49) for a chosen direction $(\vartheta, \varphi) \rightarrow e(\eta, \zeta; h)$.

2. Equate the result $e(\eta, \zeta; h)$ of step 1 with the exact ground state energy $e_{\text{top}} = -\frac{2}{3}$ of the unperturbed toric code and solve for the magnetic field strength $h \rightarrow h(\eta, \zeta)$.

3. Find the minimum $h_{\text{min}}$ of the field strength $h(\eta, \zeta)$ of step 2 w. r. t. $\eta, \zeta$ in the region $0 \leq \eta \leq 1 \wedge 0 \leq \zeta \leq 1 \rightarrow (h_x^c, h_y^c, h_z^c)$.

4. Choose another direction $(\vartheta, \varphi)$ and redo steps 1 to 4, until you have acquired the desired number of points to approximate the general phase diagram.

For a given magnetic field direction this amounts to searching for the minimal field strength $h_{\text{min}}$ where the variational energy in the thermodynamic limit for some parameters $\eta_0, \zeta_0$ crosses the exact energy $e_{\text{top}} = -\frac{2}{3}$. Another possible procedure, which has not been implemented, would be to perform instead of steps 2, 3 and 4:

2'. Evaluate the result $e(\eta, \zeta; h)$ of step 1 at a chosen field strength $h \rightarrow e(\eta, \zeta)$.

3'. Find the minimum of the result $e(\eta, \zeta)$ of step 2' in the region $0 \leq \eta, \zeta \leq 1 \rightarrow e_{\text{min}}$.

4'. Redo steps 2' and 3' until you find the minimal field strength where the found minimum $e_{\text{min}}$ crosses the exact energy $e_{\text{top}} = -\frac{2}{3} \rightarrow (h_x^c, h_y^c, h_z^c)$.

The consequence would be a considerably higher computational effort, inversely proportional to the distance between two neighboring grid points of field strength values for which the minimization would be performed. Additionally, this distance would bound the precision of the critical field strength values.

## 5 Quantum phase diagram

In this section we aim at approximating the quantum phase diagram of the 3D toric code in an arbitrary uniform magnetic field. To this end we use the QP-properties deduced from the pCUT series in SubSect. 2.2, the exact dualities of Sect. 3, and the variational treatment presented in Sect. 4. Altogether, the results of this post allow a qualitive understanding and coherent picture of the quantum criticality of the 3D toric code in a uniform magnetic field. Since the variational calculation plays a crucial role, we state the variational ansatz (33) for the ground-state wave function of the 3D toric code in a field discussed in Sect. 4 again:

$$|\alpha, \beta\rangle := \mathcal{N}(\alpha, \beta) \prod_s (\mathbb{1} + \alpha A_s) \prod_p (\mathbb{1} + \beta B_p) |\vec{h}\vec{h} \ldots \vec{h}\rangle, \qquad \alpha, \beta \in [0, 1].$$

Following the numerical scheme outlined in Sect. 4, the minimization of the variational energy, see Eq. (48), determines the associated quantum phase diagram shown in Fig. 12.

Let us first interpret the different results for the three single-field cases. For $\vec{h} = (0, 0, h_z)$, the exact duality transformation of Eq. (27) implies a second-order quantum phase transition in the $(3 + 1)D$ Ising* universality class at $h_z^c \approx 0.097$ with mean-field critical exponents. The variational calculation slightly underestimates the critical point, but agrees with the second-order nature of the phase transition. By construction, the expansion of the variational energy

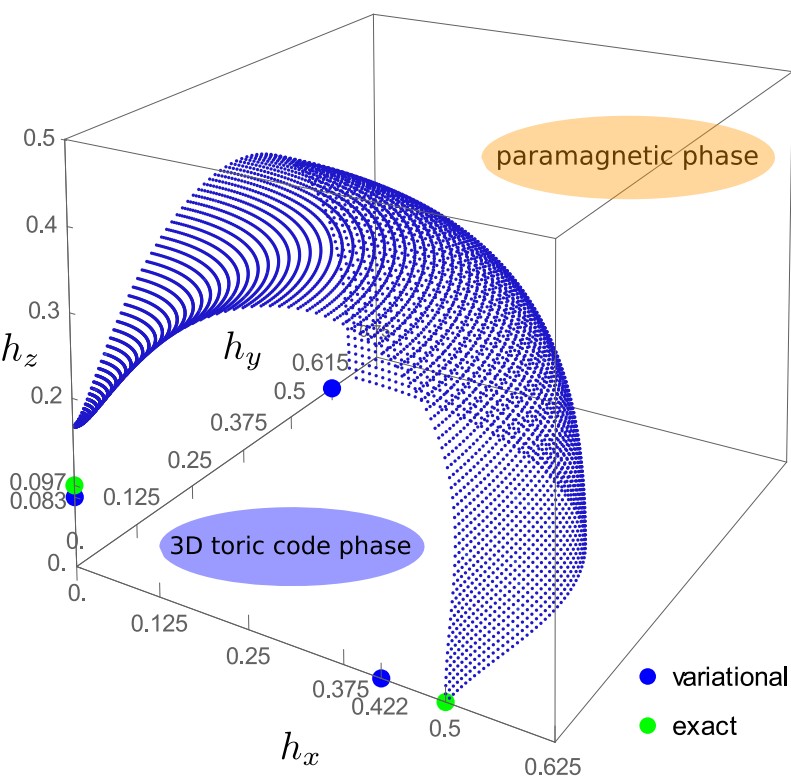

Figure 12: Quantum phase diagram of the 3D toric code in a uniform magnetic field. Blue dots represent the results of the expansion of the variational energy. Blue (green) circles give the variational (exact) critical points in the single-field cases.

leads to a first-order phase transition and yields an overestimated value for the critical point, since less quantum fluctuations are taken into account for the polarized phase. The other two single-field cases are known (or expected) to feature first-order phase transitions, which is confirmed by the variational calculation. The expansion of the variational energy is therefore expected to be a valid approximation for these cases.

Next we discuss the general case. The results of the expansion of the variational energy are shown as dots in Fig. 12. The variational parameters $\eta_c = 2\alpha_c/(1+\alpha_c^2)$ and $\zeta_c = 2\beta_c/(1+\beta_c^2)$ equal at most 0.003 at the displayed critical points. The expansion of the variational energy in these parameters appears therefore to be self-consistent. Since we see no reason for second-order phase transitions in the $h_x$-$h_y$-plane, it is reasonable that the quantum phase diagram of the 3D toric code contains a surface of first-order phase transitions for $h_z$ not too large. In contrast, at larger $h_z$ one expects a surface of second-order phase transitions in the $(3+1)D$ Ising* universality class, including the $h_z$-field case. Obviously, the expansion of the variational energy cannot capture this surface of second-order phase transitions, which is most likely similarly flat as the analogue surface of the 2D toric code [31]. Instead, the expansion gives a too-strongly bended surface. This seems to be confirmed by the approximative quantum phase diagram for the 2D toric code presented in Fig. 14 of App. E, resulting from the application of the methods of Sect. 3 and Sect. 4 to the 2D case. This phase diagram shows the same phenomenon of a too-strongly bended surface around the $h_z$-field case. The intersection of the two surfaces in the phase diagram for the 3D toric code is a line of second-order phase transitions expected to be in the $(3+1)D$ tricritical Ising* universality class.

This discussion fits in well with the results of pCUT in SubSect. 2.2 for the qualitative QP-dynamics: for $h_z$ not too large, the closing of the energy gap between the ground state and the lowest excited states causing the quantum phase transition is determined by the attrac-

tive interactions and fluctuations of immobile $m$-QP (mobile $4m$-loops cost too much energy). This hints at the first-order quantum phase transitions apparent in the approximative phase diagram. At larger $h_z$, the mobile $e$-QP rather than the immobile $m$-QP drive the quantum phase transitions by lowering their energy due to delocalization in some kind of Bose-Einstein condensation. This mechanism suggests the presence of second-order quantum phase transitions in the phase diagram. The magnetic field values where the elementary energy gaps of the $1e$-sector, $\epsilon^{(2)}_{1e,h_z,\Gamma}$, and of the $1m$-sector, $E^{(2)}_{1m}$, equal each other is determined approximately to second order in the perturbations by Eq. (21) in SubSect. 2.2; let us state it here again:

$$\epsilon^{(2)}_{1e,h_z,\Gamma} = 1 - 6h_z - 12h_z^2 \overset{!}{=} E^{(2)}_{1m} = 1 - h_x^2 - \frac{h_y^2}{3}.$$

This surface roughly indicates regions of first- and second-order phase transitions in the phase diagram. In the case of the 2D toric code in a uniform magnetic field, the qualitative picture of the QP-dynamics according to pCUT also agrees well with the approximative phase diagram in Fig. 14 of App. E and with the different refined numerical results of [31].

The approximative phase diagrams for the 2D and the 3D toric code share the features that the expansion slightly overestimates the toric code phase while the full variational ansatz in the limiting cases slightly underestimates it. Furthermore, both show dips and too-strongly bended surfaces around the cases of parallel magnetic fields. In both phase diagrams, regions of first- and second-order phase transitions cannot be distinguished by the values for the variational parameters at the phase transition points. Still, we think that the results for the 3D version are most reliable near the $h_x$-$h_y$-plane due to the indications by pCUT.

The phase diagram for the perturbed 3D toric code differs qualitatively in that it is not symmetric with respect to interchanging $h_x$ and $h_z$. This is reasonable, as in the 3D version in contrast to the 2D case, the star and plaquette operators differ from each other, since the former are 6-spin and the latter 4-spin interactions.

We can conclude that the comparison to the perturbed 2D toric code confirms the use of the expansion of the variational energy to determine approximative phase diagrams.

# 6 Conclusions and outlook

First, let us summarize the methods and results of this post and draw conclusions; secondly, an outlook in the form of promising future steps will be given.

This post can be condensed to three main messages: (1) the combination of perturbative continuous unitary transformations (pCUT) up to second order, exact duality relations and the perturbative and non-perturbative variational calculations in this post yields a reliable approximative quantum phase diagram of the toric code, a paradigmatic model of intrinsic topological order, perturbed by a uniform magnetic field. (2) The perturbed 3D toric code is robust and features a rich phase diagram (see Fig. 12), which can be qualitatively explained and consistently interpreted in the following way: (3) for the breakdown of the intrinsic topological order of the 3D toric code, the mobility of the point-like excitation, the $e$-quasiparticle (QP), and the immobility of the single constituents of spatially extended excitations, the $m$-QP, – leading to second- and first-order phase transitions, respectively – are crucial in contrast to their exotic mutual statistics.

The latter result was obtained in Sect. 2.2: first we applied pCUT to the perturbed 3D toric code in order to determine low-energy effective Hamiltonians for sectors of few interacting dressed QP, see Tab. 2, and then diagonalized them in infinite systems or by exact diagonalization in finite systems. This revealed that the change in the ground-state energy and the QP-dynamics can be understood in terms of vacuum fluctuations, hopping of $e$-QP, immobility

of single $m$-QP in all orders of the perturbations due to superselection rules, deconfinement of spatially extended excitations like $4m$-loops and short-ranged attractive interactions between QP leading to bound states between $m$-QP, as summarized in Tab. 3. Compared to the former processes, the exotic mutual statistics between $e$-QP and $m$-loops turned out to be irrelevant for the phase transitions and thus the robustness of the 3D toric code, as the statistics is only relevant in sectors with relatively large excitation energies and in relatively high orders of the perturbation. Based on these insights, we conjectured that in regions of the phase diagram where the kinetic energy of the $e$-QP dominates over the attractive interaction of the $m$-pairs, second-order phase transitions via a kind of Bose-Einstein condensation occur, while in regions where the situation is reversed, first-order phase transitions via nucleation take place. This conjecture was affirmed in the subsequent sections, as stated in main result (1) above.

In Sect. 3, the conjecture was confirmed for certain magnetic field configurations $\vec{h}$ and is consistent with the final phase diagram (Fig. 12). In these cases, duality transformations can be applied to the perturbed 3D toric code. For $\vec{h} = (h_x, 0, 0)$, the Hamiltonian is self-dual and features a first-order phase transition at exactly $h_x^c = 0.5$; for $\vec{h} = (0, 0, h_z)$, it can be mapped to the 3D transverse-field Ising model, which is known to feature a second-order phase transition at $h_z^c \approx 0.097$. The question remains open, whether the dual model for $\vec{h} = (0, h_y, 0)$, which is to the best of our knowledge novel, can be related to a known model.

Sect. 4 employed a variational ansatz for the ground state of the perturbed 3D toric code in order to determine the complete phase diagram. The ansatz was chosen such that it interpolates between the two limiting cases of the perturbed 3D toric code – the topological loop/membrane soup and the trivial polarized state. Physically, the ansatz introduces an energy cost proportional to the length of strings and surface of membranes. In order to approximate the ground states and the phase transition points, the variational energy in the thermodynamic limit had to be computed and minimized. This was possible in the cases discussed in Sect. 3 without further approximations in SubSect. 4.1 and in the general case by applying a novel expansion of the variational energy up to second order in the variational parameters $\eta$ and $\zeta$ in SubSect. 4.2. This expansion is justified when the variational parameters change rapidly at the phase transition points from small to large values, as for example for first-order phase transitions in the $h_x$-$h_y$-plane, but not for the second-order phase transition at $\vec{h} = (0, 0, h_z)$. It turned out that the first-order and the second-order expansion yield identical variational energies.

Finally, Sect. 5 combined all the insights of the previous sections to a rich phase diagram in Fig. 12 which shows that the perturbed 3D toric code is robust (main message (2) above). In this phase diagram, the exact results for the three single-field cases according to dualities, variational calculations and expansion agree qualitatively and the quantitative differences can be explained. For the general case of an arbitrary uniform magnetic field, we argued that the surface of first-order phase transitions at small $h_z$ is determined well by the expansion of SubSect. 4.2; in contrast at large $h_z$ this expansion results in a surface which seems to be bended too strongly and cannot capture the expected second-order nature of the phase transitions, as is indicated by comparing it to the 2D toric code. The phase diagram was interpreted qualitatively in terms of pCUT: depending on the strength of $h_z$, either the mobile $e$-QP (at large $h_z$) or the immobile $m$-QP (at small $h_z$) drive the second-order or first-order phase transitions, respectively. The comparison to the perturbed 2D toric code, whose phase diagram shares key properties with the more refined phase diagram in the literature [31] and the 3D case, indicated that it is valid to use the expansion of the variational energy to determine approximative phase diagrams. In conclusion, Sect. 5 showed all three main messages stated above.

How can one obtain more accurate results in the future? Obviously, the results of pCUT could be improved quantitatively by computing the effects of perturbations in orders higher

than the second. But this is a non-trivial task, because the number of terms to be calculated is relatively large due to the three perturbation parameters and three dimensions of the toric code and it increases expontially with the considered order. So computer aid is needed and a white-graph expansion [94] would probably be useful. Still, pCUT is one of the few numerical methods which can be successfully and efficiently applied to 3D systems. If one could achieve high orders, one could not only determine the dynamics of the quasiparticles in the topological and paramagnetic phase more quantitatively (so far the latter has not been investigated), but one could also locate the second-order phase transitions via a Padé extrapolation analysis [95] of the series expansions of the energy gaps. The same technique can be used to determine high-field expansions about the polarized phase, in order to pinpoint first-order phase transitions by comparing the ground-state energies of both expansions.

In contrast, the expansion of the variational energy does not seem to be improvable by simply calculating higher orders, as discussed in this post. Nevertheless, one could improve the variational results by computing the variational energy of the full ansatz numerically. This could also be used to check the calculations of the variational energy in the special cases. All these improvements are limited by the fact that apart from the two exact limiting cases the variational ansatz (33) used in this post is only an approximation to the exact ground state for a finite magnetic field strength. Hence how can this approximation be (systematically) improved? How can the quality of the approximation be assessed? The framework of tensor networks, for example in the flavor of variational PEPS, provides answers to both questions, since the variational ansatz can be represented as tensor network state (PEPS) in several ways, e.g., as a suitable 3D version of the double-line tensor network used in [93] for the 2D toric code in a magnetic field $\vec{h} = (h_x, 0, 0)$. Increasing the bond dimension and thus the number of variational parameters of the chosen tensor network state systematically improves the approximation of the ground-state wave function and energy; quantifying the convergence of this energy with increasing bond dimension enables one to assess the quality of the approximation. On the contrary, the advantage of the variational methods of this post over tensor network approaches is that the computations are much easier due to less variational parameters. One further promising route is to combine perturbative expansions and iPEPS calculations as recently realized for the 2D toric code in a field [34].

Beside the robustness and phase diagram of the toric code in a uniform magnetic field, those of other models with similar structures could potentially be investigated using the combination of pCUT, duality relations and variational methods as in this post; examples include the 3D string-net models [65], the 3D double-semion model [69,96] and certain exactly soluble models of fracton topological order [58–63] like Haah's code [60] and the X-Cube model [63]. Fracton phases have come into the focus of research recently, since despite their translational invariance they feature immobile (confined) elementary excitations due to superselection sectors. This resembles the $m$-QP of the 3D toric code. Additionally, if fracton phases are realizable, they might be employed as thermally stable, self-correcting, fault-tolerant topologically-protected quantum memories. To this end it is essential to investigate how robust they are against ubiquitous perturbations (quantum fluctuations) like a uniform magnetic field. Our conjecture is that due to the immobility of the fractons Haah's code in such a magnetic field features first-order phase transitions like the 3D toric code.

# Acknowledgements

While working on this paper, DAR was financially supported by the Deutsche Forschungsgesellschaft within the Collaborative Research Center TRR 227 "Ultrafast Spintronics" and by the Max Weber Program in the Elite Network of Bavaria. We thank L. Balents for fruitful discus-

sions.

## A  Implications of self-duality for perturbative expansions

Mentioned in Sect. 3, here we state the necessary conditions – implied by self-duality of $H(\lambda)$, $\lambda \in [0, \infty)$ – for the $n$-th order coefficients $E_{>/<}^{(n)}$ of perturbative expansions around the limits $\lambda_0^{-1} = 0$ and $\lambda_0 = 0$, respectively, assuming that $H(\lambda)$ features one phase transition at $\lambda_c$:

$$\text{for } \lambda < \lambda_c : \qquad E(\lambda) = E_<^{(0)} + E_<^{(1)}\lambda + E_<^{(2)}\lambda^2 + \mathcal{O}(\lambda^3),$$

$$\overset{\text{self-dual}}{\Rightarrow} \quad \text{for } \lambda > \lambda_c : \qquad E(\lambda) \overset{!}{=} \lambda\left( E_<^{(0)} + E_<^{(1)}\frac{1}{\lambda} + E_<^{(2)}\frac{1}{\lambda^2} + \mathcal{O}\left(\frac{1}{\lambda^3}\right)\right). \tag{50}$$

Perturbation theory in $\lambda^{-1}$ around the limit $\lambda_0^{-1} = 0$ yields

$$\text{for } \lambda > \lambda_c : \qquad E_>(\lambda) = \lambda\left( E_>^{(0)} + E_>^{(1)}\frac{1}{\lambda} + E_>^{(2)}\frac{1}{\lambda^2} + \mathcal{O}\left(\frac{1}{\lambda^3}\right)\right). \tag{51}$$

In summary self-duality implies for the expansion coefficients that

$$E_<^{(0)} = E_>^{(0)}, \; E_<^{(1)} = E_>^{(1)}, \; E_<^{(2)} = E_>^{(2)}, \; \dots . \tag{52}$$

## B  Calculation of the variational energy for the configuration $\vec{h} = (0, 0, h_z)$

As supplement to Subsubsect. 4.1.1, the calculation of the variational energy for the ansatz (37) and $\vec{h} = (0, 0, h_z)$ will be presented in the following. Analogous to [81], the first step is to compute the normalization $\mathcal{N}(\alpha)$ of the wave function and in the second step the expectation value of the Hamiltonian for the ansatz. The definition $\eta := \frac{2\alpha}{1+\alpha^2}$ will be convenient:

$$1 \overset{!}{=} \langle\alpha|\alpha\rangle = \mathcal{N}^2(\alpha)(1+\alpha^2)^N \langle\Uparrow| \prod_s (\mathbb{1} + \eta A_s)|\Uparrow\rangle = \mathcal{N}^2(\alpha)(1+\alpha^2)^N \langle\Uparrow|\mathbb{1}|\Uparrow\rangle,$$

$$\Leftrightarrow \qquad \mathcal{N}(\alpha) = \frac{1}{(1+\alpha^2)^{N/2}}, \tag{53}$$

where $N$ is the number of stars, going to infinity in the thermodynamic limit; using that all $A_s$ commute with each other and $A_s^2 = \mathbb{1}$. The third equality holds because in the product of all stars $s$, only the term of order $\eta^0$ is proportional to the identity operator or Pauli matrices $\sigma^z$. All other terms are zero due to orthogonality. The evaluation of the expectation value of the Hamiltonian yields:

$$a_s(\alpha) := \langle\alpha|A_s|\alpha\rangle = \mathcal{N}^2(\alpha)(1+\alpha^2)^N \langle\Uparrow|(\eta\mathbb{1} + A_s)\prod_{s'\neq s}(\mathbb{1} + \eta A_{s'})|\Uparrow\rangle \overset{(53)}{=} \eta,$$

$$b_p := \langle\alpha|B_p|\alpha\rangle = 1,$$

$$s_j(\alpha) := \langle\alpha|\sigma_j^z|\alpha\rangle = \mathcal{N}^2(\alpha)(1+\alpha^2)^{N-2}(1-\alpha^2)^2 \langle\Uparrow|\sigma_j^z \prod_{s'\neq s_1,s_2}(\mathbb{1} + \eta A_{s'})|\Uparrow\rangle =$$

$$\overset{(53)}{=} \left(\frac{1-\alpha^2}{1+\alpha^2}\right)^2 = 1 - \eta^2, \tag{54}$$

$$\Rightarrow \quad e(\alpha) \equiv e(\alpha, \beta = 1) := \frac{\langle\alpha|H_{\text{eff}}^z|\alpha\rangle}{3N} = -\left(\frac{2\alpha}{3\cdot 2(1+\alpha^2)} + \frac{1}{2}\right) - h_z\left(\frac{1-\alpha^2}{1+\alpha^2}\right)^2,$$

where again only the term of order $\eta^1$ contributes to the expectation value $a_s(\alpha)$, the anticommutation relation of Pauli matrices was used, $s_1$ and $s_2$ denote the stars containing spin $j$ and $e(\alpha)$ is the variational energy per spin.

## C  Transformations between the representations of the Pauli matrices and their eigenstates in the $\sigma^z$-basis and rotated basis

For the variational methods applied in SubSect. 4.2 to the general case $h_x \neq 0, h_y \neq 0, h_z \neq 0$, it is convenient to work in the basis $\{|\vec{h}\rangle, |-\vec{h}\rangle\}$. In order to do so, the transformations of Pauli matrices and eigenstates between the commonly used basis $\{|\uparrow\rangle, |\downarrow\rangle\}$ and the new basis are needed, for example expressed in the spherical coordinates of the Bloch sphere: the eigenstates of the operator $\vec{h} \cdot \vec{\sigma}$ are given by

$$
\begin{aligned}
|\vartheta, \varphi\rangle \equiv |\vec{h}\rangle &:= \cos\left(\frac{\vartheta}{2}\right)|\uparrow\rangle + e^{i\varphi}\sin\left(\frac{\vartheta}{2}\right)|\downarrow\rangle, \\
|\pi - \vartheta, \varphi + \pi\rangle \equiv |-\vec{h}\rangle &= \sin\left(\frac{\vartheta}{2}\right)|\uparrow\rangle - e^{i\varphi}\cos\left(\frac{\vartheta}{2}\right)|\downarrow\rangle \\
\text{for} \quad h_x &= |\vec{h}|\sin(\vartheta)\cos(\varphi), \quad h_y = |\vec{h}|\sin(\vartheta)\sin(\varphi), \quad h_z = |\vec{h}|\cos(\vartheta).
\end{aligned}
\tag{55}
$$

Then the Pauli matrices can be expressed in the new basis as

$$
\{|\vec{h}\rangle, |-\vec{h}\rangle\} \triangleq \left\{\begin{pmatrix} 1 \\ 0 \end{pmatrix}, \begin{pmatrix} 0 \\ 1 \end{pmatrix}\right\} \quad \Rightarrow \quad \sigma^x \triangleq \begin{pmatrix} \langle\vec{h}|\sigma^x|\vec{h}\rangle & \langle\vec{h}|\sigma^x|-\vec{h}\rangle \\ \langle-\vec{h}|\sigma^x|\vec{h}\rangle & \langle-\vec{h}|\sigma^x|-\vec{h}\rangle \end{pmatrix}, \sigma^y \triangleq \ldots, \sigma^z \triangleq \ldots,
$$

which evaluates to

$$
\sigma^x \triangleq \begin{pmatrix} \sin(\vartheta)\cos(\varphi) & -\cos(\vartheta)\cos(\varphi) - i\sin(\varphi) \\ -\cos(\vartheta)\cos(\varphi) + i\sin(\varphi) & -\sin(\vartheta)\cos(\varphi) \end{pmatrix},
\tag{56}
$$

$$
\sigma^y \triangleq \begin{pmatrix} \cos(\vartheta)\sin(\varphi) & -\cos(\vartheta)\sin(\varphi) + i\cos(\varphi) \\ -\cos(\vartheta)\sin(\varphi) - i\cos(\varphi) & -\cos(\vartheta)\sin(\varphi) \end{pmatrix},
\tag{57}
$$

$$
\sigma^z \triangleq \begin{pmatrix} \cos(\vartheta) & \sin(\vartheta) \\ \sin(\vartheta) & -\cos(\vartheta) \end{pmatrix}.
\tag{58}
$$

## D  Calculation of the variational energy for the general configuration $\vec{h} = (h_x, h_y, h_z)$

In this appendix, the calculations for the expansion of the variational energy used in SubSect. 4.2 are summarized. The following Tab. 5, 6, 7, 8, 9 and 10 contain the relevant terms in the normalization $\mathcal{N}^2$ of the variational ansatz (33) and the ansatz' expectation values for the star operators $A_s$, plaquette operators $B_p$ and Pauli matrices $\sigma_i^x, \sigma_i^y$ and $\sigma_i^z$. We again use the abbreviations $x := \sin(\vartheta)\cos(\varphi)$, $y := \cos(\vartheta)\sin(\varphi)$ and $z := \cos(\vartheta)$.

The first part of the entries in the tables up to the second double line contain the zeroth- and first-order terms in parts already shown in SubSect. 4.2. In the case of the normalization, the second part displays all second-order terms. The number of possible configurations of star, plaquette and spin operators leading to different terms increase rapidly with order, see Fig. 13. Therefore, for the expectation values the second part of the entries contains only the relevant terms for the general case $h_x \neq 0, h_y \neq 0, h_z \neq 0$ (in short $h_x, h_y, h_z$) in the thermodynamic limit $N \to \infty$. These terms result from configurations of *unconnected* stars

Table 4: List of abbreviations used in Tab. 5, 6, 7, 8, 9 and 10. In all cases, "next neighbor" means the direct neighbor of the direct neighbor.

| quantity | abbreviation |
|---|---|
| ratio no. stars to no. of unit cells | $r$ |
| ratio no. stars to no. of spins | $r_s$ |
| ratio no. plaquettes to no. of spins | $r_p$ |
| no. spins/stars in/neighboring star | $n_s$ |
| no. spins in plaquette | $n_p$ |
| no. plaquettes neighboring star | $n_{ps}$ |
| no. stars neighboring plaquette | $n_{sp}$ |
| no. plaquettes neighboring plaquette | $n_{pp}$ |
| no. next neighbor stars to star | $n_{n,ss}$ |
| no. next neighbor plaquettes to star | $n_{n,ps}$ |
| no. next neighbor stars to plaquette | $n_{n,sp}$ |
| no. next neighbor plaquettes to plaquette | $n_{n,pp}$ |
| no. stars neighboring spin | $\bar{n}_s$ |
| no. plaquettes neighboring spin | $\bar{n}_p$ |
| no. next neighbor stars to spin | $\bar{n}_{n,s}$ |
| no. next neighbor plaquettes to spin | $\bar{n}_{n,p}$ |

and plaquettes. The terms' order in $N$ equals the highest considered order in $\eta$ and $\zeta$. In the special cases $(h_x,0,0),(0,h_y,0)$ and $(0,0,h_z)$, additional terms can contribute for $N \to \infty$, as terms of higher order in $N$, which dominate in the general case, can be killed. Thus the only relevant terms for these special cases for $N \to \infty$ are stated in parentheses. In the special case $(h_x,0,0)$, set the variational parameter $\eta = 1$ and for $(0,0,h_z)$ set $\zeta = 1$, like in the ansatz (40) and in (37) for the full variational calculations. Ignore all terms in the tables containing $\eta, A_s$ or $\zeta, B_p$, respectively. Finally, the expectation value of $A_s$ ($B_p$) contributes $\frac{r_s}{2}$ ($\frac{r_p}{2}$) to the variational energy.

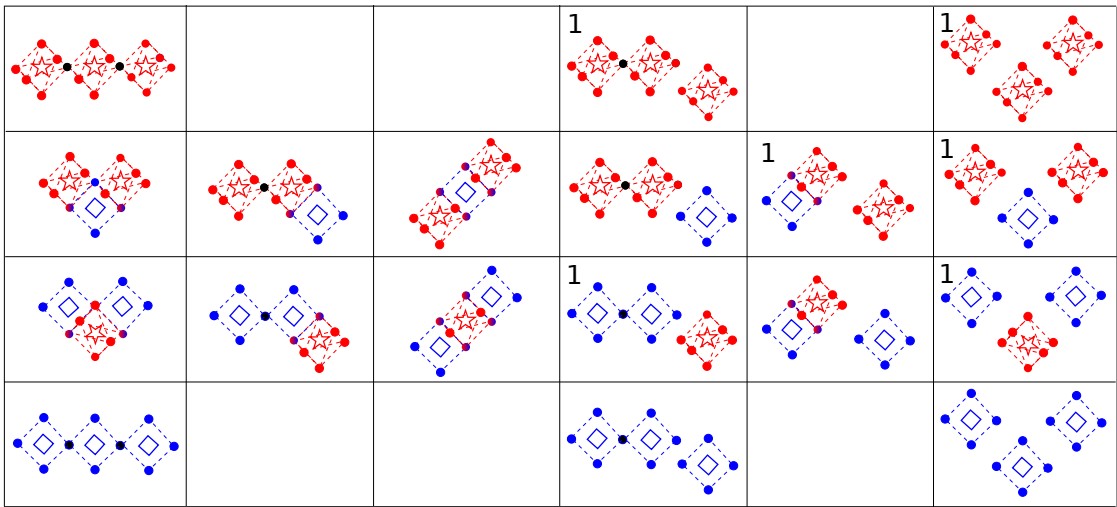

Figure 13: Configurations of connected and unconnected star and plaquette operators. Configuration $(i;j)$ refers to the $i$th row and $j$th column; $^1(i;j)$ encodes configuration $(i;j)$ without the unconnected star.

Table 5: List of terms calculated for the normalization up to second order in $\eta$ and $\zeta$, as explained in the main text. The notation $(\neg) < i, j >$ encodes that star or plaquette $i$ (does not) neighbors $j$. For the notation see also Fig. 13 and Tab. 4.

| NORMALIZATION | | | |
|---|---|---|---|
| order: configuration | diagram | exact value | value for $N \to \infty$ |
| $\eta^0 \zeta^0$: $\mathbb{1}$ | - | $+1$ | $0$ ($+1$ for $(0, h_y, 0)$) |
| $\eta^1$: $\sum_s A_s$ | - | $N \cdot x^{n_s}$ | $0$ |
| $\zeta^1$: $\sum_p B_p$ | - | $rN \cdot z^{n_p}$ | $0$ |
| $\eta^2$: $\sum_s \sum_{s', s' \neq s} A_s A_{s'};\quad < s, s' >$ | $^1(1;4)$ | $N n_s \cdot x^{2n_s - 2}$ | $0$ |
| $\eta^2$: $\sum_s \sum_{s', s' \neq s} A_s A_{s'};\quad \neg < s, s' >$ | $^1(1;6)$ | $N(N - n_s - 1) \cdot x^{2n_s}$ | $N^2 x^{2n_s} \eta^2$ |
| $\eta^1 \zeta^1$: $\sum_s \sum_p A_s B_p;\quad < s, p >$ | $^1(2;5)$ | $-N n_{ps} \cdot x^{n_s - 2} y^2 z^{n_p - 2}$ | $0$ |
| $\eta^1 \zeta^1$: $\sum_s \sum_p A_s B_p;\quad \neg < s, p >$ | $^1(2;6)$ | $N(rN - n_{ps}) \cdot x^{n_s} z^{n_p}$ | $rN^2 x^{n_s} z^{n_p} \eta \zeta$ |
| $\zeta^2$: $\sum_p \sum_{p', p' \neq p} B_p B_{p'};\quad < p, p' >$ | $^1(3;4)$ | $rN n_{pp} \cdot z^{2n_p - 2}$ | $0$ |
| $\zeta^2$: $\sum_p \sum_{p', p' \neq p} B_p B_{p'};\quad \neg < p, p' >$ | $^1(3;6)$ | $rN(rN - n_{pp} - 1) \cdot z^{2n_p}$ | $r^2 N^2 z^{2n_p} \zeta^2$ |

Table 6: List of terms calculated for the star operator up to second order in $\eta$ and $\zeta$, as explained in the main text. For the notation see also Fig. 13, Tab. 4 and Tab. 5.

| EXPECTATION VALUE OF $A_s$ | | | |
|---|---|---|---|
| order: configuration | diagram | exact value | value for $N \to \infty$ |
| $\eta^0 \zeta^0$: $A_s$ | - | $x^{n_s}$ | $0$ |
| $\eta^1$: $\mathbb{1}$ | - | $+1$ | $0$ ($\eta$ for $(0, h_y, 0)$) |
| $\eta^1$: $A_s \sum_{s', s' \neq s} A_{s'};\quad < s, s' >$ | $^1(1;4)$ | $n_s \cdot x^{2n_s - 2}$ | $0$ |
| $\eta^1$: $A_s \sum_{s', s' \neq s} A_{s'};\quad \neg < s, s' >$ | $^1(1;6)$ | $(N - n_s - 1) \cdot x^{2n_s}$ | $0$ |
| $\zeta^1$: $A_s \sum_p B_p;\quad < s, p >$ | $^1(2;5)$ | $-n_{ps} \cdot x^{n_s - 2} y^2 z^{n_p - 2}$ | $0$ |
| $\zeta^1$: $A_s \sum_p B_p;\quad \neg < s, p >$ | $^1(2;6)$ | $(rN - n_{ps}) \cdot x^{n_s} z^{n_p}$ | $0$ |
| $\eta^2$: $A_s \sum_{s1} \sum_{s2, s2 \neq s1} A_{s1} A_{s2};$ $\neg < s, s_1 >, \neg < s, s_2 >,$ $\neg < s_1, s_2 >$ | $(1;6)$ | $\left[ (N - n_{n,ss} - n_s - 1)(N - 2n_s - 2) + n_{n,ss}(N - 2n_s - 1) \right] \cdot x^{3n_s}$ | $N^2 x^{3n_s} \eta^2$ |
| $\eta^1 \zeta^1$: $A_s \sum_p \sum_{s', s' \neq s} B_p A_{s'};$ $\neg < s, p >, \neg < s', p >,$ $\neg < s, s' >$ | $(2;6)$ | $\left[ (rN - n_{n,ps} - n_{ps})(N - n_{sp} - n_s - 1) + n_{n,ps}(N - n_{sp} - n_s) \right] \cdot x^{2n_s} z^{n_p}$ | $rN^2 x^{2n_s} z^{n_p} \eta \zeta$ |
| $\zeta^2$: $A_s \sum_p \sum_{p', p' \neq p} B_p B_{p'};$ $\neg < s, p >, \neg < s, p' >,$ $\neg < p, p' >$ | $(3;6)$ | $\left[ (rN - n_{n,ps} - n_{ps})(rN - n_{ps} - n_{pp} - 1) + n_{n,ps}(rN - n_{ps} - n_{pp}) \right] \cdot x^{n_s} z^{2n_p}$ | $r^2 N^2 x^{n_s} z^{2n_p} \zeta^2$ |

Table 7: List of terms calculated for the plaquette operator up to second order in $\eta$ and $\zeta$, as explained in the main text. For the notation see also Fig. 13, Tab. 4 and Tab. 5.

| EXPECTATION VALUE OF $B_p$ | | | |
|---|---|---|---|
| order: configuration | diagram | exact value | value for $N \to \infty$ |
| $\eta^0\zeta^0$: $B_p$ | - | $z^{n_p}$ | 0 |
| $\eta^1$: $B_p \sum_s A_s$; $\quad <s,p>$ | $^1(2;5)$ | $-n_{sp} \cdot x^{n_s-2} y^2 z^{n_p-2}$ | 0 |
| $\eta^1$: $B_p \sum_s A_s$; $\quad \neg <s,p>$ | $^1(2;6)$ | $(N-n_{sp}) \cdot x^{n_s} z^{n_p}$ | 0 |
| $\zeta^1$: $\mathbb{1}$ | - | $+1$ | 0 ($\zeta$ for $(0,h_y,0)$) |
| $\zeta^1$: $B_p \sum_{p',p'\neq p} B_{p'}$; $\quad <p,p'>$ | $^1(3;4)$ | $n_{pp} \cdot z^{2n_p-2}$ | 0 |
| $\zeta^1$: $B_p \sum_{p',p'\neq p} B_{p'}$; $\quad \neg <p,p'>$ | $^1(3;6)$ | $(rN-n_{pp}-1) \cdot z^{2n_p}$ | 0 |
| $\eta^2$: $B_p \sum_s \sum_{s',s'\neq s} A_s A_{s'}$; <br> $\neg <s,p>, \neg <s',p>, \neg <s,s'>$ | $(2;6)$ | $\ldots$ | $N^2 x^{2n_s} z^{n_p} \eta^2$ |
| $\eta^1\zeta^1$: $B_p \sum_s \sum_{p'} A_s B_{p'}$; <br> $\neg <s,p>, \neg <s,p'>, \neg <p,p'>$ | $(3;6)$ | $\ldots$ | $rN^2 x^{n_s} z^{2n_p} \eta\zeta$ |
| $\zeta^2$: $B_p \sum_{p1} \sum_{p2,p2\neq p1} B_{p1} B_{p2}$; <br> $\neg <p,p_1>, \neg <p,p_2>, \neg <p_1,p_2>$ | $(4;6)$ | $\ldots$ | $r^2 N^2 z^{3n_p} \zeta^2$ |

Table 8: List of terms calculated for the Pauli matrix $\sigma_i^x$ up to second order in $\eta$ and $\zeta$, as explained in the main text. For the notation see also Fig. 13, Tab. 4 and Tab. 5.

| EXPECTATION VALUE OF $\sigma_i^x$ | | | |
|---|---|---|---|
| order: configuration | diagram | exact value | value for $N \to \infty$ |
| $\eta^0\zeta^0$: $\sigma_i^x$ | - | $x$ | 0 ($x$ for $(h_x,0,0)$) |
| $\eta^1$: $\sigma_i^x \sum_s A_s$; $\quad <i,s>$ | - | $\bar{n}_s \cdot x^{n_s-1}$ | 0 |
| $\eta^1$: $\sigma_i^x \sum_s A_s$; $\quad \neg <i,s>$ | - | $(N-\bar{n}_s) \cdot x^{n_s+1}$ | 0 |
| $\zeta^1$: $\sigma_i^x \sum_{p,\neg <i,p>} B_p$; $\quad \neg <i,p>$ | - | $(rN-\bar{n}_p) \cdot x z^{n_p}$ | 0 |
| $\eta^2$: $\sigma_i^x \sum_s \sum_{s',s'\neq s} A_s A_{s'}$; <br> $\neg <i,s>, \neg <i,s'>, \neg <s,s'>$ | $^1(1;6)$ + spin | $\ldots$ | $N^2 x^{2n_s+1} \eta^2$ |
| $\eta^1\zeta^1$: $\sigma_i^x \sum_s \sum_{p,\neg <i,p>} A_s B_p$; <br> $\neg <i,s>, \neg <i,p>, \neg <s,p>$ | $^1(2;6)$ + spin | $\ldots$ | $rN^2 x^{n_s+1} z^{n_p} \eta\zeta$ |
| $\zeta^2$: $\sigma_i^x \sum_p \sum_{p',p'\neq p,\neg <i,p'>} B_p B_{p'}$; <br> $\neg <i,p>, \neg <i,p'>, \neg <p,p'>$ | $^1(3;6)$ + spin | $\ldots$ | $r^2 N^2 x z^{2n_p} \zeta^2$ |

Table 9: List of terms calculated for the Pauli matrix $\sigma_i^y$ up to second order in $\eta$ and $\zeta$, as explained in the main text. For the notation see also Fig. 13, Tab. 4 and Tab. 5.

| EXPECTATION VALUE OF $\sigma_i^y$ | | | |
|---|---|---|---|
| order: configuration | diagram | exact value | value for $N \to \infty$ |
| $\eta^0\zeta^0$: $\sigma_i^y$ | - | $y$ | $0$ ($y$ for $(0, h_y, 0)$) |
| $\eta^1$: $\sigma_i^y \sum_s A_s$; $\quad \neg <i,s>$ | - | $(N - \bar{n}_s) \cdot x^{n_s} y$ | $0$ |
| $\zeta^1$: $\sigma_i^y \sum_{p, \neg <i,p>} B_p$; $\quad \neg <i,p>$ | - | $(rN - \bar{n}_p) \cdot yz^{n_p}$ | $0$ |
| $\eta^2$: $\sigma_i^y \sum_s \sum_{s', s' \neq s, \neg <i,s'>} A_s A_{s'}$; $\neg <i,s>, \neg <i,s'>, \neg <s,s'>$ | $^1(1;6)$ + spin | | $N^2 x^{2n_s} y \eta^2$ |
| $\eta^1\zeta^1$: $\sigma_i^y \sum_{s, \neg <i,s>} \sum_{p, \neg <i,p>} A_s B_p$; $\neg <i,s>, \neg <i,p>, \neg <s,p>$ | $^1(2;6)$ + spin | | $rN^2 x^{n_s} yz^{n_p} \eta\zeta$ |
| $\zeta^2$: $\sigma_i^y \sum_p \sum_{p', p' \neq p, \neg <i,p'>} B_p B_{p'}$; $\neg <i,p>, \neg <i,p'>, \neg <p,p'>$ | $^1(3;6)$ + spin | | $r^2 N^2 yz^{2n_p} \zeta^2$ |

Table 10: List of terms calculated for the Pauli matrix $\sigma_i^z$ up to second order in $\eta$ and $\zeta$, as explained in the main text. For the notation see also Fig. 13, Tab. 4 and Tab. 5.

| EXPECTATION VALUE OF $\sigma_i^z$ | | | |
|---|---|---|---|
| order: configuration | diagram | exact value | value for $N \to \infty$ |
| $\eta^0\zeta^0$: $\sigma_i^z$ | - | $z$ | $0$ ($z$ for $(0, 0, h_z)$) |
| $\eta^1$: $\sigma_i^z \sum_s A_s$; $\quad \neg <i,s>$ | - | $(N - \bar{n}_s) \cdot x^{n_s} z$ | $0$ |
| $\zeta^1$: $\sigma_i^z \sum_{p, \neg <i,p>} B_p$; $\quad <i,p>$ | - | $\bar{n}_p \cdot z^{n_p - 1}$ | $0$ |
| $\zeta^1$: $\sigma_i^z \sum_{p, \neg <i,p>} B_p$; $\quad \neg <i,p>$ | - | $(rN - \bar{n}_p) \cdot z^{n_p + 1}$ | $0$ |
| $\eta^2$: $\sigma_i^z \sum_s \sum_{s', s' \neq s, \neg <i,s'>} A_s A_{s'}$; $\neg <i,s>, \neg <i,s'>, \neg <s,s'>$ | $^1(1;6)$ + spin | | $N^2 x^{2n_s} z \eta^2$ |
| $\eta^1\zeta^1$: $\sigma_i^z \sum_{s, \neg <i,s>} \sum_p A_s B_p$; $\neg <s,p>, \neg <i,s>, \neg <i,p>$ | $^1(2;6)$ + spin | | $rN^2 x^{n_s} z^{n_p + 1} \eta\zeta$ |
| $\zeta^2$: $\sigma_i^z \sum_p \sum_{p', p' \neq p} B_p B_{p'}$; $\neg <i,p>, \neg <i,p'>, \neg <p,p'>$ | $^1(3;6)$ + spin | | $r^2 N^2 z^{2n_p + 1} \zeta^2$ |

# E Approximative quantum phase diagram for the 2D toric code in a uniform magnetic field

Fig. 14 presents the approximative quantum phase diagram for the 2D toric code in a uniform magnetic field resulting from the application of the methods of Sect. 3 and Sect. 4.

SciPost Phys. **6**, 078 (2019)

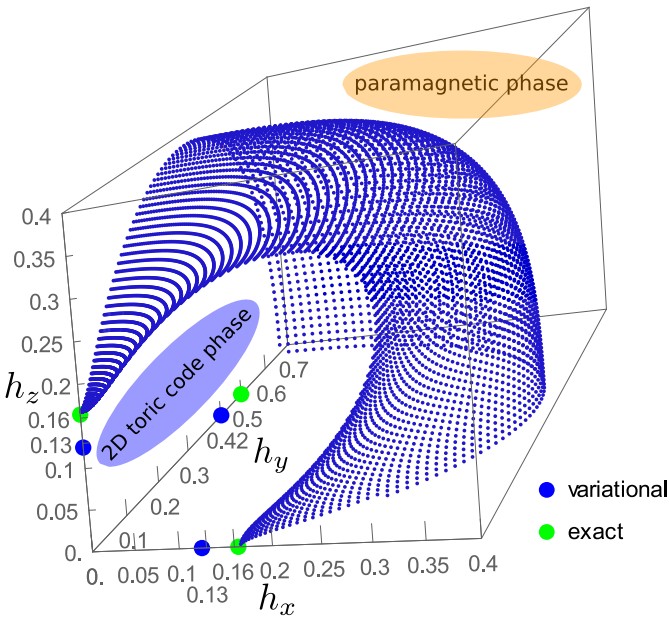

Figure 14: Quantum phase diagram of the 2D toric code in a uniform magnetic field. Blue dots represent the results of the expansion of the variational energy. Blue (green) circles give the variational (exact) critical points in the single-field cases.

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
