# Peer review of "Quantum robustness and phase transitions of the 3D Toric Code in a field"

_SciPost Physics, doi:SciPost Phys. 6, 078 (2019)_

## Round 1 · Referee Report · Irénée Frerot · 2019-4-18

Strengths

1- The introduction offers a very useful entry to the literature on the 2d and 3d toric code, as well as related models and implementations.
2- The properties of the unperturbed 3d toric code on a cubic lattice are well exposed, in a manner accessible to the newcomer.
3- The phase diagram under an external field is identified consistently through a variety of methods (exactly in certain limits, and by variational methods).
4- Throughout the manuscript, the authors try to provide a physical intuition of the mechanisms at play, as well as comparisons with the better-known 2d toric code.

Weaknesses

1- The technicalities related to the variational determination of the ground state are hard to follow.

Report

In the paper, the authors study the 3d toric code on a cubic lattice, which displays a topologically ordered ground state, and focus on the phase transitions towards a trivial state induced by an external uniform magnetic field. After a detailed introduction which clearly motivates their study, the authors describe the model and its ground state properties. Then, applying a combination of methods (p-CUT, exact dualities and variational computations), they reconstruct the phase diagram of the toric code under a uniform magnetic field in an arbitrary direction. Throughout this study, they provide a detailed description of the physical mechanisms induced by the external field, and compare them with the 2d case. Finally, a long discussion summarizes the main results of the paper.

Overall, I find the paper extremely well written, and accessible to the newcomer to the field. Although the technicalities associated with pCUT and the variational calculations are difficult to follow, their outputs are clearly summarized in physical terms. The phase diagram is convincingly reconstructed via various independent means. For these reasons, I recommend the publication of the manuscript.

Requested changes

1- Eq. (6) should be explained a bit more. The authors should explain why the ground-state degeneracy is given by $2^{N_{spins}} / 2^{N_{constraints}}$.
2- I think that Eq. (8) contains a typo. First, I would advise the authors not to use m as a variable inside the summation, as it brings confusion with the superscript m in $P^m_{xy}$. Then, I think that a $b_z$ is missing in front of the term $(n_z + 1/2)$.
3- Eq. (9) contains a typo : the last term is $b_z = (0, 0, 1)$.
4- p.7, first line: I would suggest to add "In the loop-soup picture of the ground state, these operators measure the parity..."
5- Just after Eq. (11): "with some fixed $(n_y, n_z) \in Z$..."
6- Footnote 3: "...can also be viewed, in the light of quantum codes, as the..." (with comas)
7- After Eq. (18): is really the ground state energy equal to 4N ? I would say that $E_0 = -(1/2)N_{stars} -(1/2)N_{plaquettes} = -N/2 - 3N/2 = -2N$, am I correct?
8- Eq. (19): what means the prefactor $1 / (2j)$ in front of the last term?
9- Eq. (21) and in other places, the authors use the symbol "=" with a "!" on top of it: I have never seen this symbol and some explanation would be welcome.
10- After Eq. (33): "...the two limiting cases $\alpha=\beta=1$ and $\alpha=\beta=0$ are exactly..." and "For $\alpha=\beta=1$, the normalization..."

  • validity: high
  • significance: high
  • originality: good
  • clarity: high
  • formatting: excellent
  • grammar: perfect

Author:  Kai Phillip Schmidt  on 2019-05-27  [id 528]

(in reply to Report 1 by Irénée Frerot on 2019-04-18)

Dear Irénée Frerot,

we thank you for carefully examining our paper and for the overall very positive comments on our work. In the revised version of our article, we have addressed the minor issues raised in your report.

To be specific,

1) In the revised we have explained Eq. 6 in more detail. 2) We agree and we have updated Eq. 8. 3) We agree and we have updated Eq. 9. 4) Here we do not agree. Our statement is not only valid in the loop-soup picture of the ground state. So we have left the formulation as it is. 5) We agree and we have "n_x \in Z" as suggested by the referee. 6) We agree and we haved added the commas as suggested by the referee. 7) We agree with the referee and updated the formula. 8) The factor 1/(2j) has to be inside the sum. We corrected this. 9) We have added a footnote on page 15 to explain this symbol. 10) We have followed the referee and changed the expressions.

Additionally, we added one page 17 the reference

M.H. Zarei, Physical Review B 96, 165146 (2019).

for the dual Hamiltonian of the 3D TFIM.

Best regards,

David A. Reiss Kai P. Schmidt.

---

## Round 2 · Author Response

We thank the referee for carefully examining our article and for the overall
very positive comments on our work. In the revised version, we have addressed
the minor issues raised in the referee report and we have added one more reference.

---

## Round 2 · List of Changes

In the revised version of our article, we have addressed the following minor issues raised in the referee's report:

1) In the revised version we have explained Eq. 6 in more detail.
2) We have updated Eq. 8 according to the referees suggestions.
3) We have updated Eq. 9 according to the referees suggestions.
4) Here we do not agree with the referee. Our statement is not only valid in the loop-soup picture of the ground state. So we have left the formulation as it is.
5) We agree and we have "n_x \in Z" as suggested by the referee.
6) We agree and we haved added the commas as suggested by the referee.
7) We agree with the referee and updated the formula.
8) The factor 1/(2j) in Eq. 19 has been put inside the sum. We corrected this.
9) We have added a footnote on page 15 to explain this symbol.
10) We have followed the referee and changed the expressions on page 19.

Additionally, we added on page 17 the reference
M.H. Zarei, Physical Review B 96, 165146 (2019).
for the dual Hamiltonian of the 3D TFIM.

---

## Editorial Decision

published